# CAS: A Probability-Based Approach for Universal Condition Alignment Score

**Chunsan Hong**[*]
KAIST

South Korea
hoarer@kaist.ac.kr

**ByungHee Cha**[*]
Seoul National Univ.

South Korea
paulcha1025@snu.ac.kr

**Tae-Hyun Oh**[†]
Dept. EE and GSAI, POSTECH &
Inst. for CREAT, Yonsei Univ.
South Korea
taehyun@postech.ac.kr

## Abstract

Recent conditional diffusion models have shown remarkable advancements and have been widely applied in fascinating real-world applications. However, samples generated by these models often do not strictly comply with user-provided conditions. Due to this, there have been few attempts to evaluate this alignment via pre-trained scoring models to select well-generated samples. Nonetheless, current studies are confined to the text-to-image domain and require large training datasets. This suggests that crafting alignment scores for various conditions will demand considerable resources in the future. In this context, we introduce a universal condition alignment score that leverages the conditional probability measurable through the diffusion process. Our technique operates across all conditions and requires no additional models beyond the diffusion model used for generation, effectively enabling self-rejection. Our experiments validate that our metric effectively applies in diverse conditional generations, such as text-to-image, {instruction, image}-to-image, edge-/scribble-to-image, and text-to-audio. Project page: https://unified-metric.github.io/

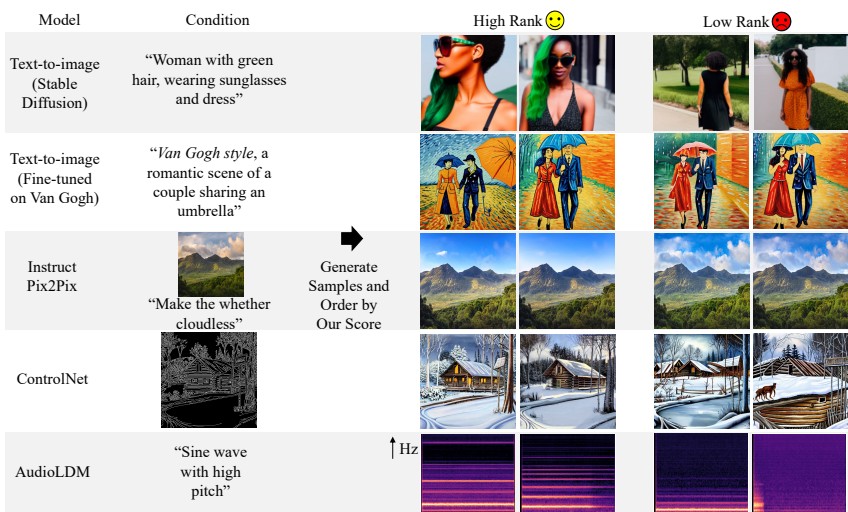

Figure 1: Results of generated samples ordered by our universal score for various conditions. From left to right, the sequence shows the conditional diffusion model used, the employed condition, two high/low-ranked samples. The last column's generation is audio, so it is visualized using a Mel Spectrogram. Our metric demonstrates automatic and reliable assessment of perceptual alignment between condition and generated results across text-to-image models (Rombach et al., 2022), diffusion models trained on specific domains such as Van Gogh, InstructPix2Pix (Brooks et al., 2023), ControlNet (Zhang et al., 2023), and AudioLDM (Liu et al., 2023), but not limited to.

---

[*]These authors contributed equally to this work.
[†]Corresponding author.

# 1 INTRODUCTION

The recent surge of large-scale diffusion-based text-to-image models, *e.g.*, (Saharia et al., 2022; Brooks et al., 2023; Zhang et al., 2023; Rombach et al., 2022; Yang et al., 2023; Ramesh et al., 2022), has drawn widespread attention to the overall conditional diffusion models. The conditional diffusion models take arbitrary conditional inputs provided by the users and generate impressive samples conforming the given conditions. While text-to-image is the most popular example, the modalities and types of conditions are not limited, *e.g.*, image-to-image (Brooks et al., 2023), pose-/segmentation-to-image (Zhang et al., 2023), text-to-audio (Liu et al., 2023), *etc*. These developments exhibit impressive compositional generalization abilities and complying behaviors according to unbounded conditions. Despite those impressive quality, diffusion models generate undesirable samples that are not perfectly aligned with or even further neglect the given conditions. In this case, the common practice is to generate many samples for a given condition and manually pick preferred images among the diverse samples. This demands human's effort, cost and time.

To mitigate these costs and automate the processes, in the specific text-to-image (T2I) application, numerous learning-based alignment scores (Hessel et al., 2021; Wu et al., 2023; Xu et al., 2023; Kirstain et al., 2023) have been proposed, where they train a separate neural network model to predict a perceptual alignment score between a text and an image. Likewise, there are a few attempts to define individual scores for each problem depending on different forms of the condition and the output, *e.g.*, Gal et al. (2022) present the directional CLIP similarity for image editing.

The previous development methods are all anchored for each specific format pair of condition, input, and output. These require to design alignment scores individually for various conditions, such as text, segmentation, pose, and others. Establishing a new alignment score for each of the countless types of condition pairs requires an immense amount of effort. In particular, learning-based approaches, *e.g.*, (Hessel et al., 2021; Wu et al., 2023; Xu et al., 2023; Kirstain et al., 2023), strongly rely on extensive and well-curated data for training neural models, which demands significant expenditures in terms of both cost and time for data collection and model training. Furthermore, there are practical scenarios, where collecting data and training a new score model are even more challenging; for example, specifically to the T2I field, conditional diffusion models are often fine-tuned for various domains for user-specific needs, such as animation, pixel art, or even the distinctive style of a particular artist. However, our experimental findings suggest that existing metrics, relying on pre-trained models, markedly underperform in these scenarios.

In this regard, we propose Condition Alignment Score (CAS), the universal score to evaluate alignment with the given condition and generated samples, *regardless of the type of conditions and without any separate model training and data*. Our key idea is to use the conditional likelihood $p(x|c)$ which can be obtained from the diffusion model as CAS. We utilize the same diffusion model which is used in generating samples without modification, *i.e.* self-rejection, thereby addressing all the previously mentioned shortcomings. By leveraging the diffusion model itself, no pre-trained scoring model is needed, and our technique possesses the capability to reject out-of-distribution images from the domain in which the diffusion was trained. Since the conditional probability itself is used as metric, there is no need to design separate scores for each condition individually.

We leverage the exact probability computation method by Song et al. (2021b), yet the conditional probability is not measurable during the sampling process where Classifier Free Guidance (CFG) (Ho & Salimans, 2022) is employed. Therefore we utilize the inversion process. However, we observe that the common practice of the inversion method (Song et al., 2021a), called Denoising Diffusion Implicit Model (DDIM) inversion, does not provide accurate alignment of the reverse process due to the utilization of approximations. We additionally present a recursive inversion method which improves the performance of our method. Our main contributions are summarized as follows:

- We present a universal alignment score applicable to all possible conditions.

- Our self-rejection technique leverages the diffusion model that is used for generating images, thereby eliminating the dependency on well-curated datasets and separately pre-trained score models that the previous studies grappled with.

- We developed a method for improved DDIM inversion, boosting the performance of our technique.

## 2 RELATED WORK

Since our scope is related to the assessment of the alignment between given conditions and generated samples, we first brief the evaluation conventions used in the most popular application of diffusion, text-to-image (T2I). Among T2I diffusion models, Imagen (Saharia et al., 2022), Stable Diffusion (Rombach et al., 2022), and DALL-E2 (Ramesh et al., 2022) stand out as the most prominent. In these studies, the primary metrics proposed for T2I alignment largely rely on human evaluation and the CLIP Score (Hessel et al., 2021). CLIP Score measures the cosine similarity between the text representation and the generated image representation obtained from the pre-trained CLIP model (Radford et al., 2021). The most recent studies (Wu et al., 2023; Kirstain et al., 2023; Xu et al., 2023) propose learning-based metrics in which training data includes prompts, multiple images generated from each prompt, and their human preference annotations. All the aforementioned scores rely on learning models pre-trained on elaborated or vast datasets. Beyond the T2I diffusion models, various kinds of conditional diffusion models such as ControlNet (Zhang et al., 2023), InstructPix2Pix (Brooks et al., 2023), and AudioLDM (Liu et al., 2023) have been proposed with different combinations of modalities for conditions and generated targets. Considering vast application scenarios, it would be impractical to define and develop respective scores for each combination case; thus, it has been barely studied for general use cases.

Our work focuses on measuring the conditional probability of generated samples, and for this, we employ the method proposed by Song et al. (2021b). They present a method to measure the log-likelihood of images through a pre-trained diffusion model, which has been adapted in various studies (Zimmermann et al., 2021; Feng et al., 2023). Zimmermann et al. (2021) utilize this method to measure the class-conditioned likelihood of images on CIFAR-10 for classification, achieving promising results. However, we find that calculating conditional probability in this manner does not work as desired in complex conditions, such as natural language prompts. Our approach diverges in that we additionally exploit unconditional probability for debiasing, thereby expanding the effectiveness to numerous conditions. Feng et al. (2023) utilize their method to solve the inverse problem and empirically exhibit that the image prior is measured close to the real-world prior. However, while they aim to solve problems using the prior, our approach differs in that we strive to solve problems leveraging the likelihood.

Research measuring the likelihood probability is not only confined to diffusion models, but has been explored in other generative models, such as flow-based generative models. Flow-based generative models (Dinh et al., 2015; Rezende & Mohamed, 2015; Kingma & Dhariwal, 2018) come with a method to calculate the likelihood of a sample and utilize it as an objective function in model training. Nevertheless, this approach bears a critical drawback: it imposes substantial constraints on the architecture of models to satisfy the required flow properties. This limitation hampered the development of flow-based generative models for practical applications, which is why we utilize diffusion models. Nonetheless, this remains a notable area, as the method proposed by Song et al. (2021b) is inspired by the formula that flow-based generative models aim to solve.

## 3 PRELIMINARY

**DDIM and DDIM inversion.** Denoising Diffusion Probabilistic Model (DDPM) (Ho et al., 2020) serves as generative models that aim to approximate real data distribution $q(\mathbf{x})$ by a model $p_{\boldsymbol{\theta}}(\mathbf{x})$. For each $\mathbf{x}_0 \sim q(\mathbf{x})$, DDPM constructs a discrete Markov chain $\{\mathbf{x}_0, \mathbf{x}_1, ..., \mathbf{x}_N\}$ that satisfies $p(\mathbf{x}_t|\mathbf{x}_{t-1}) = N(\mathbf{x}_t; \sqrt{1 - \beta_t}\mathbf{x}_{t-1}, \beta_t\mathbf{I})$. This process is termed the forward step, where the sequence $\{\beta_t\}_{t=1}^N$ is a sequence of positive noise scales, *i.e.*, $0 < \beta_1, \beta_2, ..., \beta_N < 1$. Assuming that $N$ is considerably large, DDPM presumes that $p_N(\mathbf{x}) \approx N(0; \mathbf{I})$. DDPM starts by sampling $\mathbf{x}_N$ from a Gaussian distribution and then undergoes a stochastic reverse process to generate the sample $\mathbf{x}_0$, *e.g.*, image.

On the contrary, the Denoising Diffusion Implicit Model (DDIM) adapts the formula to facilitate a deterministic reverse process, enabling inversion. Given the conditions of the forward step, the following equation is satisfied: $\mathbf{x}_t = \sqrt{\alpha_t}\mathbf{x}_0 + \sqrt{1 - \alpha_t}\boldsymbol{\epsilon}, \boldsymbol{\epsilon} \sim N(0; \mathbf{I})$, where $\alpha_t = \prod_{i=1}^t (1 - \beta_i)$. By training the diffusion model $\boldsymbol{\theta}$ to predict noise $\boldsymbol{\epsilon}$ that is deviated to the original image $\mathbf{x}_0$ with given $\mathbf{x}_t$, *i.e.* training with the objective function $\mathbb{E}[||\boldsymbol{\epsilon}_{\boldsymbol{\theta}}(\mathbf{x}_t, t) - \boldsymbol{\epsilon}||^2]$, we can perform the following sampling to generate image:

$$\mathbf{x}_{t-1} = \sqrt{\alpha_{t-1}}(\frac{\mathbf{x}_t - \sqrt{1-\alpha_t}\boldsymbol{\epsilon_\theta}(\mathbf{x}_t, t)}{\sqrt{\alpha_t}}) + \sqrt{1 - \alpha_{t-1} - \sigma_t^2}\boldsymbol{\epsilon_\theta}(\mathbf{x}_t, t) + \sigma_t\boldsymbol{\epsilon}_t. \tag{1}$$

Setting $\sigma_t = 0$ for all $t$ results in a deterministic DDIM sampling process. When rearranging this equation for $\mathbf{x}_t$, it becomes the ideal DDIM inversion, which recovers the latent from the image:

$$\mathbf{x}_t = a_{t-1}\mathbf{x}_{t-1} + b_{t-1}\boldsymbol{\epsilon_\theta}(\mathbf{x}_t, t), \tag{2}$$

where $a_{t-1} = \frac{\sqrt{\alpha_t}}{\sqrt{\alpha_{t-1}}}$, $b_{t-1} = (\sqrt{1-\alpha_t} - \sqrt{\frac{\alpha_t}{\alpha_{t-1}} - \alpha_t})$, and $\sigma_t = 0$. However, it is contradictory to use $x_t$ itself to recover $x_t$. Therefore, it is convention to approximate it by:

$$\hat{\mathbf{x}}_t = a_{t-1}\mathbf{x}_{t-1} + b_{t-1}\boldsymbol{\epsilon_\theta}(\mathbf{x}_{t-1}, t), \tag{3}$$

so that $\mathbf{x}_t \approx \hat{\mathbf{x}}_t$. This is referred to as the DDIM inversion. The effectiveness of this approximation hinges on the linearity assumption to the latents, which is proved by Miyake et al. (2023).

**Probability Flow ODE and Exact Probability Computation.** Song et al. (2021b) present probability flow ordinary differential equation (ODE). Contrary to methods like DDIM and DDPM, where $t$ is treated as a discrete variable, in probability flow ODE, $t$ is considered as a continuous variable, where latent is denoted as $\mathbf{x}(t)$. We change the domain of $t$ from $[1, N]$ to $[0, 1]$ for probability flow ODE. Then, probability flow ODE is given as follows: $d\mathbf{x} = \mathbf{f_\theta}(\mathbf{x}(t), t)dt$. By applying the instantaneous change of variable formula provided by Chen et al. (2018) to this equation, one can compute the probability value of the synthesized sample $\mathbf{x}(0)$:

$$\log p_0(\mathbf{x}(0)) = \log p_1(\mathbf{x}(1)) + \int_0^1 \nabla_\mathbf{x} \cdot \mathbf{f_\theta}(\mathbf{x}(t), t)dt, \tag{4}$$

where $p_1(\mathbf{x}) \approx N(0; \mathbf{I})$. Since computation of $\nabla_\mathbf{x} \cdot \mathbf{f_\theta}(\mathbf{x}, t)$ is intractable, Song et al. (2021b) approximately compute this value by Skilling-Hutchinson trace estimator (Skilling, 1989; Hutchinson, 1989): $\nabla_\mathbf{x} \cdot \mathbf{f_\theta}(\mathbf{x}, t) \approx \mathbb{E}_{p(\mathbf{z})}[\mathbf{z}^\top \nabla_\mathbf{x}\mathbf{f_\theta}(x, t)\mathbf{z}]$. $\mathbf{z}$ is a random variable which satisfies $\mathbb{E}_{p(\mathbf{z})}[\mathbf{z}] = 0$ and $\text{Cov}_{p(\mathbf{z})}[\mathbf{z}] = \mathbf{I}$, and $\nabla_\mathbf{x}\mathbf{f_\theta}$ is Jacobian matrix of $\mathbf{f_\theta}$ where $\mathbf{z}^\top \nabla_\mathbf{x}\mathbf{f_\theta}(\mathbf{x}, t)$ can be computed by reverse mode automatic differentiation.

**Classifier Free Guidance (CFG).** When training a conditional diffusion model, learning solely with $\mathbb{E}[||\boldsymbol{\epsilon_\theta}(\mathbf{x}_t, \mathbf{c}, t) - \boldsymbol{\epsilon}||^2]$ does not yield satisfactory results. Thus, CFG (Ho & Salimans, 2022) is commonly employed. In this method, both $\mathbb{E}[||\boldsymbol{\epsilon_\theta}(\mathbf{x}_t, \emptyset, t) - \boldsymbol{\epsilon}||^2]$ and $\mathbb{E}[||\boldsymbol{\epsilon_\theta}(\mathbf{x}_t, \mathbf{c}, t) - \boldsymbol{\epsilon}||^2]$ are simultaneously trained within a single diffusion network, and an image is sampled through the following equation: $\tilde{\boldsymbol{\epsilon}}_{\boldsymbol{\theta}}(\mathbf{x}, \mathbf{c}, t) = \boldsymbol{\epsilon_\theta}(\mathbf{x}, \emptyset, t) + w \cdot (\boldsymbol{\epsilon_\theta}(\mathbf{x}, \mathbf{c}, t) - \boldsymbol{\epsilon_\theta}(\mathbf{x}, \emptyset, t))$. We will denote the latent variable derived through CFG as $\{\tilde{\mathbf{x}}_t\}_{t=0}^N$.

## 4 METHOD

### 4.1 METRIC DEFINITION

Our goal is to estimate the true conditional likelihood $\log p(\mathbf{x}|\mathbf{c})$ of the generated sample $\mathbf{x}$ given condition $\mathbf{c}$ as a fundamental metric for CAS. In this section, we delve into the rationale behind the decision of our score formulation defined by probability. The methodology for calculating probability will be discussed further in Sec. 4.2. We approximate the conditional likelihood $p_{\boldsymbol{\theta}}(\mathbf{x}|\mathbf{c}) \approx p(\mathbf{x}|\mathbf{c})$, by the diffusion model as a competent metric for measuring whether generated samples satisfy the given condition. However, upon ranking with this metric, we discovered that it did not exhibit meaningful tendencies. This lack of a meaningful pattern would be primarily due to the diffusion model's inability to mimic the true likelihood accurately.

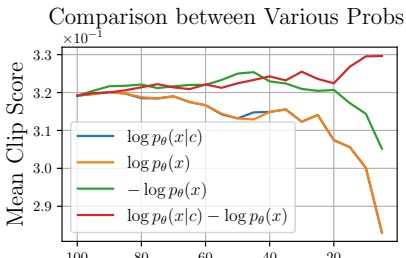

Figure 2: Average CLIP Score (Hessel et al., 2021) of top N% images cherry-picked by each metric.

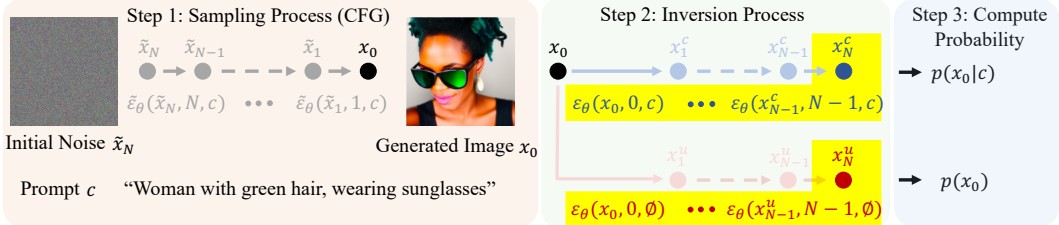

Figure 3: The figure illustrates our method and which latents of the diffusion process are used to compute CAS, $\log p_{\boldsymbol{\theta}}(\mathbf{x}_0|\mathbf{c}) - \log p_{\boldsymbol{\theta}}(\mathbf{x}_0)$. The values highlighted in yellow in Step 2 are needed. Therefore, to measure CAS, the image first undergoes an inversion process (Step 2), and then the values obtained during this process are used to calculate $p_{\boldsymbol{\theta}}(\mathbf{x}_0)$ and $p_{\boldsymbol{\theta}}(\mathbf{x}_0|\mathbf{c})$ (Step 3).

In our analysis, we found a significant bias stemming from $p_{\boldsymbol{\theta}}(\mathbf{x})$ when calculating $p_{\boldsymbol{\theta}}(\mathbf{x}|\mathbf{c})$. We conducted a preliminary experiment using the Stable Diffusion v1.5 model Rombach et al. (2022) in Fig. 2. We synthesized 100 images from the prompt "Woman with green hair, wearing sunglasses and dresses." We then compared the average CLIP Score (Hessel et al., 2021) of top N% images cherry-picked by each of $\log p_{\boldsymbol{\theta}}(\mathbf{x})$, $-\log p_{\boldsymbol{\theta}}(\mathbf{x})$, $\log p_{\boldsymbol{\theta}}(\mathbf{x}|\mathbf{c})$ and $\log p_{\boldsymbol{\theta}}(\mathbf{x}|\mathbf{c}) - \log p_{\boldsymbol{\theta}}(\mathbf{x})$. Fig. 2 illustrates that images boasting a higher $\log p_{\boldsymbol{\theta}}(\mathbf{x}|\mathbf{c})$ tend to possess a markedly lower CLIP Score, and that the tendency of $\log p_{\boldsymbol{\theta}}(\mathbf{x}|\mathbf{c})$ and that of $\log p_{\boldsymbol{\theta}}(\mathbf{x})$ are almost the same. Since a lower CLIP Score indicates that the image is not aligned with the prompt, this advocates the hypothesis that $\log p_{\boldsymbol{\theta}}(\mathbf{x}|\mathbf{c})$ does not mimic the true likelihood, $p(\mathbf{x}|\mathbf{c})$. On the other hand, we observed $-\log p_{\boldsymbol{\theta}}(\mathbf{x})$ shows better performance than $\log p_{\boldsymbol{\theta}}(\mathbf{x})$. Therefore, we venture the following hypotheses: 1) the conditions for generating images are often unseen in the training data of the diffusion model; hence resulting in a low measurement of $p(\mathbf{x})$ and 2) this effect is so pronounced that it biases $p(\mathbf{x}|\mathbf{c})$. We speculate that the task of forming the likelihood distribution of an "unseen condition" demands strong extrapolation assumptions, which in reality may not hold true. Sehwag et al. (2022) demonstrate the difficulty in probability estimation in diffusion models when faced with unseen images, while Kirichenko et al. (2020) reveal similar challenges within flow-based generative models. Based on these findings, we hypothesize that extrapolation for probability computation on unseen conditions may pose challenges in our scenario.

To address this inherent bias, it is imperative to correct the root cause, which is the skewed $\log p_{\boldsymbol{\theta}}(\mathbf{x})$. To this end, we introduce a correction factor into our ranking metric, $S = \log p_{\boldsymbol{\theta}}(\mathbf{x}|\mathbf{c}) - \lambda \log p_{\boldsymbol{\theta}}(\mathbf{x})$. In our experiments, we set $\lambda = 1$. With this setting, it is equivalent to picking a sample based on $\log p_{\boldsymbol{\theta}}(\mathbf{c}|\mathbf{x})$ where $p(\mathbf{c})$ is constant. As shown in Fig. 2, images with higher values of our metric tend to align well with conditional text description.

## 4.2 DDIM in ODE form & probability computation

Based on the idea of Sec. 4.1, we define CAS and present its computation method utilizing equation 4. We denote the latent generated from the unconditional inversion process at time step $t$ as $\mathbf{x}_t^u$, and the latent generated from the conditional inversion process at time step $t$ as $\mathbf{x}_t^c$.

Both $p_{\boldsymbol{\theta}}(\mathbf{x})$ and $p_{\boldsymbol{\theta}}(\mathbf{x}|\mathbf{c})$ cannot be measured in the reverse process since CFG breaks the stochastic modeling of the diffusion model. Instead, the probabilities can be obtained during the inversion process which is the analogous process to recovering latents. In Fig. 3, we illustrate the process of how we obtain the latents necessary for probability computation from conditional diffusion models. When $N$ is large, by transferring the time domain from $[1, N]$ to $[0, 1]$, we can convert $\mathbf{x}_t^u$ and $\mathbf{x}_t^c$ into continuous variables with respect to t as $\mathbf{x}^u(t/N)$ and $\mathbf{x}^c(t/N)$. Accordingly, CAS is defined:

**Definition 1** (Condition Alignment Score (CAS))**.** *Given diffusion model, $\boldsymbol{\theta}$, the latent at time $t$ from an unconditional DDIM inversion process initiated from an image $\mathbf{x}(0)$, $\mathbf{x}^u(t)$, and the corresponding latent at the time $t$ from a conditional DDIM inversion process also beginning from $\mathbf{x}(0)$, $\mathbf{x}^c(t)$, the CAS is defined as:*

$$CAS(\mathbf{x}, \mathbf{c}, \boldsymbol{\theta}) = \log \frac{p_1(\mathbf{x}^c(1)|\mathbf{c})}{p_1(\mathbf{x}^u(1))} - \int_0^1 \alpha'(t) \frac{\nabla_{\mathbf{x}} \cdot \boldsymbol{\epsilon}_{\boldsymbol{\theta}}(\mathbf{x}^c(t), c, t) - \nabla_{\mathbf{x}} \cdot \boldsymbol{\epsilon}_{\boldsymbol{\theta}}(\mathbf{x}^u(t), \emptyset, t)}{2\alpha(t)\sqrt{1-\alpha(t)}} dt. \quad (5)$$

We can compute equation 5 using a numerical solver. To derive the above metric, we start by representing the DDIM in the form of an ordinary differential equation (ODE), (the detailed derivation is provided in Supplementary Material)

$$\frac{d\mathbf{x}}{dt} = \alpha'(t)\left(\frac{\mathbf{x}(t)}{2\alpha(t)} - \frac{\boldsymbol{\epsilon}_{\boldsymbol{\theta}}(\mathbf{x}(t),t)}{2\alpha(t)\sqrt{1-\alpha(t)}}\right). \tag{6}$$

Applying equation 4 to equation 6, we can compute the log probability in DDIM inversion process:

$$\log p_0(\mathbf{x}^u(0)) = \log p_1(\mathbf{x}^u(1)) + \log\frac{\sqrt{\alpha(0)}}{\sqrt{\alpha(1)}}D - \int_0^1 \alpha'(t)\frac{\nabla_{\mathbf{x}}\cdot\boldsymbol{\epsilon}_{\boldsymbol{\theta}}(\mathbf{x}^u(t),\emptyset,t)}{2\alpha(t)\sqrt{1-\alpha(t)}}dt, \tag{7}$$

where $D$ is the number of dimensions of $\mathbf{x}$. For a given condition $\mathbf{c}$, the conditional log probability is expressed as:

$$\log p_0(\mathbf{x}^c(0)|\mathbf{c}) = \log p_1(\mathbf{x}^c(1)|\mathbf{c}) + \log\frac{\sqrt{\alpha(0)}}{\sqrt{\alpha(1)}}D - \int_0^1 \alpha'(t)\frac{\nabla_{\mathbf{x}}\cdot\boldsymbol{\epsilon}_{\boldsymbol{\theta}}(\mathbf{x}^c(t),\mathbf{c},t)}{2\alpha(t)\sqrt{1-\alpha(t)}}dt. \tag{8}$$

By subtracting equation 7 from equation 8, we derive the formula for CAS in Definition 1, which offers a principled framework for evaluating the alignment between a sample and a given condition through calculating associated probabilities. Note that applying CAS is not limited to image domains, but can be applied to arbitrary modalities of input and condition.

## 4.3 IMPROVING DDIM INVERSION PROCESS VIA DDIM RECURSIVE INVERSION

To obtain an accurate measure of probability, the inversion process must perfectly align with the reverse process. However, the current DDIM inversion does not, as an error occurs when approximating $\mathbf{x}_t \approx \hat{\mathbf{x}}_t$ (Mokady et al., 2023; Miyake et al., 2023). Although several studies have been conducted to address this issue in image editing, Mokady et al. (2023); Miyake et al. (2023) present the compensation methods exclusively for image editing rather than accurately calculating the inversion. On the other hand, Wallace et al. (2023) defines a precise method for inversion using two coupled variables. However, it is unsuitable for probability computation, since it is necessary to solve the coupled variable differential equation. In contrast, we precisely invert the DDIM without bells and whistles. Under the simple assumption that $\hat{\mathbf{x}}_t$ is closer to $\mathbf{x}_t$ than $\mathbf{x}_{t-1}$, we can establish the following equation:

**Definition 2** (DDIM Recursive Inversion). *We define the i-th step recursive inversion as:*

$$\hat{\mathbf{x}}_t^i = a_{t-1}\mathbf{x}_{t-1} + b_{t-1}\boldsymbol{\epsilon}_{\boldsymbol{\theta}}(\hat{\mathbf{x}}_t^{i-1},t) \tag{9}$$

*for $i = 2, 3, ...$ and $\hat{\mathbf{x}}_t^1 = \hat{\mathbf{x}}_t$. We denote the approximation $\mathbf{x}_t \approx \hat{\mathbf{x}}_t^n$ as the n-th order recursive inversion.*

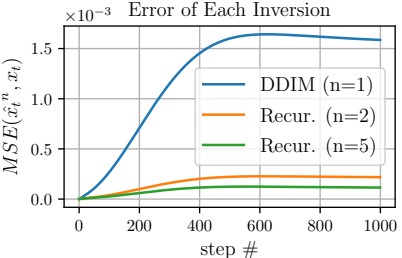

Figure 4: Mean Squared Error (MSE) of latent during the reverse process and inversion process of the original DDIM inversion and our method (recursive inversion). We sample $\mathbf{x}_N$ from a Gaussian distribution and then generate $\{\mathbf{x}_t\}_{k=0}^{N-1}$ through a reverse process. Subsequently, starting from $\mathbf{x}_0$, we implement the inversion processes to produce $\{\hat{\mathbf{x}}_t^n\}_{t=1}^N$, where setting $n = 1$ becomes original DDIM inversion and $n > 1$ becomes our recursive inversion.

The difference is apparent by comparing with the original DDIM inversion $\hat{\mathbf{x}}_t = a_{t-1}\mathbf{x}_{t-1} + b_{t-1}\boldsymbol{\epsilon}_{\boldsymbol{\theta}}(\mathbf{x}_{t-1},t)$. To verify the validity of Definition 2, we empirically test it on the CelebA dataset, as shown in Fig. 4. This preliminary test exhibits that, through a mere utilization of our proposed equation, the error between the inversion process and the reverse process can be diminished to less than 10% of the original DDIM inversion, *i.e.*, accurate. In addition, one might question whether simply increasing the number of the inversion steps could be beneficial. From our experiments, the recursive inversion not only saves much time compared to just increasing the steps but also enhances the cherry-picking performance of CAS more. We provide the evidence for these claims in Supplementary Material.

| $\sigma$ | 0.1 | 0.01 | 0.001 | 0.0001 | 0.00001 |
|---|---|---|---|---|---|
| NRMSE | 0.164 | 0.012 | 0.002 | 0.021 | 0.212 |

Table 1: Normalized Root Mean Square Error (NRMSE) between log probability computed using Song et al. (2021b) and log probability computed using our method. NRMSE is defined as $NRMSE(y, \hat{y}) = \sqrt{1/N \cdot \sum_i^N ((y_i - \hat{y}_i)/y_i)^2}$ where $y$ is ground truth and $\hat{y}$ is approximation.

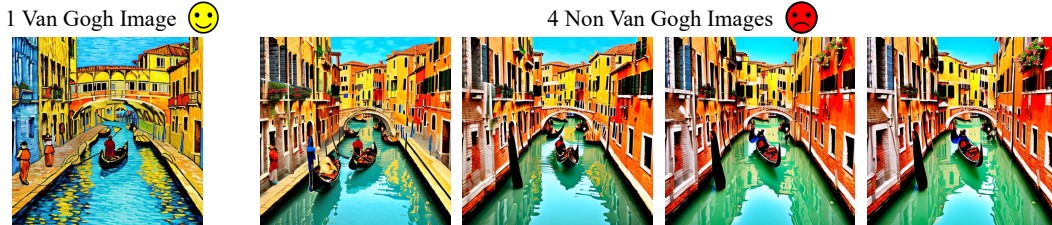

Figure 5: Example of 5 generated images from Van Gogh Diffusion. The prompt used for generating images is "Van Gogh style, a romantic gondola ride under Venetian bridges". However, except for the leftmost image, negative weighting to "Van Gogh style" was done for the rest of the images, leading to a noticeable deviation from the intended artistic style.

## 4.4 FASTER COMPUTATION

In this section, we propose an efficient and faster method to approximate Skilling-Hutchinson trace estimator (Skilling, 1989; Hutchinson, 1989) $\mathbb{E}_{p(\mathbf{z})}[\mathbf{z}^\top \nabla_\mathbf{x} \boldsymbol{\epsilon_\theta}(\mathbf{x}, t)\mathbf{z}]$. Song et al. (2021b) bypass Jacobian computation from Skilling-Hutchinson estimator by directly calculating $\mathbf{z}^\top \nabla_\mathbf{x} \boldsymbol{\epsilon_\theta}(\mathbf{x}, t)$ using backpropagation. However, backpropagation requires expensive computations. To avoid the resource-intensive backpropagation process, we have approximately computed $\mathbf{z}^\top \nabla_\mathbf{x} \boldsymbol{\epsilon_\theta}(\mathbf{x}, t)\mathbf{z}$ as:

$$\mathbf{z}^\top \nabla_\mathbf{x} \boldsymbol{\epsilon_\theta}(\mathbf{x}, t)\mathbf{z} \simeq \mathbf{z}^\top \frac{\boldsymbol{\epsilon_\theta}(\mathbf{x} + \sigma\mathbf{z}, t) - \boldsymbol{\epsilon_\theta}(\mathbf{x}, t)}{\sigma} = \frac{1}{\sigma}\mathbf{z}^\top (\boldsymbol{\epsilon_\theta}(\mathbf{x} + \sigma\mathbf{z}, t) - \boldsymbol{\epsilon_\theta}(\mathbf{x}, t)), \quad (10)$$

where $\sigma$ is a sufficiently small number. By leveraging the above expressions, we can make the approximation $\mathbb{E}_{p(\mathbf{z})}[\mathbf{z}^\top \nabla_\mathbf{x} \boldsymbol{\epsilon_\theta}(\mathbf{x}, t)\mathbf{z}]$ feasible even without resorting to backpropagation. Notably, with only two forward operations, the process becomes significantly faster, reducing the total time to 16% with negligible error. Furthermore, GPU memory used for backpropagation is also saved. In this work, we set and fix $\sigma$ to 0.001, which shows the best approximation of log probability in our empirical test as shown in Table 1.

## 5 EXPERIMENTS

In our experimental section, we validate the efficacy of our approach through tests on five distinct models: a T2I diffusion model fine-tuned in the Van Gogh domain (Mackay, 2023), Stable Diffusion v1.5 (Rombach et al., 2022), InstructPix2Pix (Brooks et al., 2023), ControlNet (Zhang et al., 2023), and AudioLDM (Liu et al., 2023). Through experiments, we validate that CAS consistently delivers high performance in various domains without necessitating additional model training.

## 5.1 T2I DIFFUSION MODEL

**Baselines.** To validate ability of CAS as T2I alignment score, we selected several baselines: CLIP Score (Hessel et al., 2021), Human Preference Score (HPS) (Wu et al., 2023), Image Reward (Xu et al., 2023), and Pick Score (Kirstain et al., 2023). CLIP Score leverages CLIP (Radford et al., 2021) model which is trained with real image and text annotation pairs. In contrast, HPS, Image Reward, and Pick Score utilize models that are trained based on human-ranked annotations of multiple images generated from text prompts. Details are shown in Table 2. Our method utilizes 2nd order recursive inversion with a total step of 10. The comparison between our method and the established baselines

| | CLIP Score | Image Reward | HPS | Pick Score | Ours |
|---|---|---|---|---|---|
| Train data (Image) | 400M | 4–9/Text | ~99K | ~584K | **None** |
| Train data (Text) | 400M | ~9K | ~25K | ~38K | **None** |
| Acc. on Van Gogh dataset | 0.284 | 0.247 | 0.229 | 0.338 | **0.470 $\pm$ 0.013** |
| Acc. on Pick Score dataset | 0.580 | 0.621 | 0.697 | **0.721** | 0.622 $\pm$ 0.004 |

Table 2: Amount of training data for each metric, and accuracy on our conducted Van Gogh dataset and Pick Score dataset. For our method, accuracy is measured five times, and we indicate the 95% confidence interval for these measurements.

| Model | InstructPix2Pix | Canny Edge ControlNet | Scribble ControlNet | AudioLDM |
|---|---|---|---|---|
| Accuracy (%) | 59.2 | 64.3 | 61.0 | 58.3 |

Table 3: Human preference evaluation on each model. The accuracy listed in the table represents the average preference accuracy across five participants.

hinges on two pivotal criteria: 1) The capacity to measure whether the image has a style that the diffusion model is fine-tuned on, and 2) The accuracy in evaluating T2I Alignment. Each of them is measured under the experiment done in our conducted Van Gogh dataset and Pick Score dataset.

**Experiment Setting in Van Gogh Dataset.** Evaluating the style fidelity in domain-specific diffusion models is important. Typically, this style is specified within the prompt. Therefore, the T2I alignment score should also be able to measure whether the image adheres to the specified style. To exemplify this, we conduct experiments on a diffusion model specifically trained on Vincent Van Gogh, the well-known artist. For evaluation, we utilize 275 prompts, randomly generated by GPT-4, encompassing landscapes, portraits, and still-life paintings. For each prompt, we generate five distinct images using the Van Gogh diffusion model, denoted as "Van Gogh style, {prompt}". Images are divided into two groups: 1) one image generated directly from the prompts themselves, and 2) four images generated by applying negative weighting (implemented by the diffuser (von Platen et al., 2022)) to the "Van Gogh style" portion of the prompt. We designated the ground truth for the first group as Van Gogh and for the second group as non-Van Gogh. Additionally, we conducted a human verification step. Examples of this can be found in Fig. 5, where all samples are clearly distinguished. We compute the proportion of numbers each metric successfully identifies one real Van Gogh style image per each prompt. The prompts used for evaluating T2I Alignment in each metric are also framed as "Van Gogh style, {prompt}".

**Experiment Setting in Pick Score dataset.** The Pick Score dataset consists of a total of 500 prompts, two image pairs generated from each prompt, and annotations based on human preferences. Among these, data entries marked as "indistinguishable" are excluded, and we analyze the accuracy of each method in the remaining 432 image pairs. We utilize Stable Diffusion v1.5 for our score.

**Results.** As shown in Table 2, our approach exhibits an accuracy that is approximately 13.2% higher than the second-best performing method, Pick Score in Van Gogh dataset. This result underscores the inherent advantage of our self-rejection technique, an exceptional capability to measure whether images are from the intended domain. The margin of superiority is already notable; however, given that Van Gogh is a globally renowned artist, it is plausible to infer that our method's performance would be even more pronounced in minor domains. We also observed that our method outperforms CLIP Score and is comparable with Image Reward in Pick Score dataset. While our accuracy might be lower than that of HPS or Pick Score, the fact that CAS outperforms the CLIP Score and is comparable with Image Reward without using any pre-trained scoring models is remarkable. Although it might not represent the state of the art in the T2I domain, the significance lies in the potential for saving time and resources in the majority of other conditional domains where scores have not yet been researched. The 2nd order recursive inversion achieved an accuracy about 2.9% higher than the original DDIM in Pick Score dataset, 1.1% higher than the original DDIM in the Van Gogh dataset.

## 5.2 ADDITIONAL MODALITIES

To corroborate the universal applicability of our formulated metric, we conduct human preference evaluations across multiple modalities including InstructPix2pix ({instruction, image}-to-image), ControlNet ({image, text}-to-image), and AudioLDM (text-to-audio).

**Human Preference Evaluation.** Given the challenge of finding quantitative metrics for evaluation, we resorted to human preference evaluation to validate our metric's efficiency. First, we conducted datasets composed solely of specific conditions, *e.g.* the dataset for ControlNet evaluation includes 100 canny edge images. We then generate 100 samples for each condition in the dataset. Subsequently, participants are given a condition and a pair of two images generated by the corresponding condition: 1) A randomly selected sample, and 2) a sample with top-1 CAS from the 180 samples. They are asked to select a sample that aligns with the condition more. Subsequently, we recorded the ratio of instances in which participants opted for the top-ranked generated sample.

**Experimental Setup in InstructPix2Pix.** We selected 180 random test cases from the Instruct-Pix2Pix dataset, each consisting of an original image and prompt for image editing. Since Instruct-Pix2Pix is a multi-conditional diffusion model, we add a slight twist to our metric: $\log p_{\boldsymbol{\theta}}(\mathbf{x}|\mathbf{c}_i, \mathbf{c}_p) - \log p_{\boldsymbol{\theta}}(\mathbf{x}|\mathbf{c}_i)$, where $\mathbf{c}_i$ is the original image and $\mathbf{c}_p$ is the editing prompt.

**Experimental Setup in ControlNet Model.** We evaluated two variants of image conditions: a scribble image for Scribble ControlNet and a canny edge image for Canny Edge ControlNet. To carry out the evaluation, we assembled 180 original images from InstructPix2pix. For each original image, we extracted both scribble and canny edge images. We used extracted image conditions as a test dataset (edited prompt was employed together to generate samples).

**Experimental Setup in AudioLDM Model.** To investigate our metric's efficacy in text-to-audio modalities, 180 prompts are generated by GPT-4 and used for human preference evaluation.

**Results.** The results in Table 3 demonstrate around or over 60% preference for our metric across modalities. Particularly for conditions like canny edge and scribble, despite their inherent ambiguity in alignment measurement, we observe a notable congruence with human perception. Additionally, the results from AudioLDM demonstrate that our approach functions effectively, even when the diffusion model output is not an image. Through these experiments, we verify that our method successfully measures alignment across any input-output pairs of the diffusion model.

**Comparison with Directional CLIP Similarity in Instruct-Pix2Pix** In the case of IntructPix2pix, we additionally compare the relation between CAS and directional CLIP Similarity. The directional CLIP similarity, introduced by Gal et al. (2022), measures the cosine similarity between the embedding differentials of original and edited images and texts in the InstructPix2Pix model. This metric is also employed for the evaluation of InstructPix2ix in Brooks et al. (2023). To compare our metric with directional CLIP similarity, we assess for average directional CLIP similarity per rank. The outcomes in Fig. 6 indicate an inverse proportionality between our metric and directional CLIP similarity, with a higher rank in our metric corresponding to greater directional CLIP similarity.

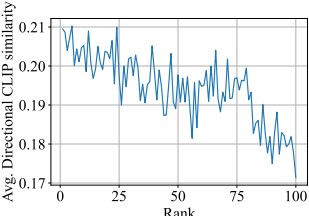

Figure 6: Average directional CLIP similarity based on CAS rank in the test data.

## 6 CONCLUSION

We have successfully introduced a CAS, that leverages conditional probability to calculate alignment between generated samples and its corresponding conditions. We find that the conditional probability $p_{\boldsymbol{\theta}}(\mathbf{x}|\mathbf{c})$ computed from the diffusion model is biased toward probability $p_{\boldsymbol{\theta}}(\mathbf{x})$. By utilizing this knowledge, we achieve success in measuring condition alignment directly from the diffusion model. Furthermore, we present the recursive inversion which reduces the error of DDIM inversion to within 10%. Additionally, by presenting an approach that approximates backpropagation computation with minimal error, we have also managed to reduce computation time to 16%. Through these methodologies, our technique demonstrates high performance across various conditions without the need for pre-trained scoring models. Our research is readily applicable across all domains utilizing the conditional diffusion model, substantially reducing the resources dedicated to cherry-picking.

ACKNOWLEDGMENT

T.-H. Oh was partially supported by Institute of Information & communications Technology Planning & Evaluation (IITP) grant funded by the Korea government (MSIT) (No.2021-0-02068, Artificial Intelligence Innovation Hub; No.RS-2023-00225630, Development of Artificial Intelligence for Text-based 3D Movie Generation; No.2022-0-00124, Development of Artificial Intelligence Technology for Self-Improving Competency-Aware Learning Capabilities) and by the National Research Foundation of Korea (NRF) grant funded by the Korea government (MSIT) (No. NRF-2021R1C1C1006799).

ETHICS STATEMENT

Generative models for synthesizing images carry with them several ethical concerns, and these concerns are shared by (or perhaps exacerbated in) any generative models such as ours. Generative models, in the hands of bad actors, could be abused to generate disinformation. Generative models such as ours may have the potential to displace creative workers via automation. That said, these tools may also enable growth and improve accessibility for the creative industry.

REPRODUCIBILITY STATEMENT

We will release the code and examples used in this work.

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

APPENDIX

|  | $\log p_{\boldsymbol{\theta}}(\boldsymbol{x}|\boldsymbol{c}_1)$ | $\log p_{\boldsymbol{\theta}}(\boldsymbol{x}|\boldsymbol{c}_2)$ | $\log p_{\boldsymbol{\theta}}(\boldsymbol{x})$ | $CAS(\boldsymbol{x},\boldsymbol{c}_1,\boldsymbol{\theta})$ | $CAS(\boldsymbol{x},\boldsymbol{c}_2,\boldsymbol{\theta})$ |
|---|---|---|---|---|---|
| $\boldsymbol{x}_1$ | 61059.39 | 61046.07 | 61011.86 | **47.53** | 34.21 |
| $\boldsymbol{x}_2$ | 59566.22 | 59612.24 | 59546.12 | 20.10 | **66.12** |

Table 4: The average $\log p_{\boldsymbol{\theta}}(\boldsymbol{x}|\boldsymbol{c})$, $\log p_{\boldsymbol{\theta}}(\boldsymbol{x})$, and $CAS(\boldsymbol{x},\boldsymbol{c},\boldsymbol{\theta})$ of each 100 images $\boldsymbol{x}_1$ and $\boldsymbol{x}_2$ generated from $\boldsymbol{c}_1 =$ "a woman with black hair" and $\boldsymbol{c}_2 =$ "a woman with rainbow hair".

| Employed $\boldsymbol{\theta}$ in $CAS(\mathbf{x},\mathbf{c},\boldsymbol{\theta})$ | Image Source | | | |
|---|---|---|---|---|
|  | DP2.0 | OJ | SD1.5 | Real |
| DP2.0 | **189.90** | 140.88 | 32.39 | 10.47 |
| OJ | 121.80 | **171.81** | 51.04 | 27.85 |
| SD1.5 | 72.50 | 54.13 | **163.24** | 36.79 |

Table 5: Average CAS values for images from multiple sources. We measured average CAS for 100 real images from the Coco dataset and 100 images each generated using Dreamlike Photoreal 2.0 (DP2.0) (Art, 2023), OpenJourney (OJ) (Prompthero, 2023), and Stable Diffusion 1.5 (SD1.5) (Rombach et al., 2022), based on 100 captions from the COCO dataset. For the CAS measurement, the same diffusion models, DP2.0, OJ, and SD1.5 are employed.

## A  FURTHER ANALYSIS OF CAS

### A.1  ADDITIONAL INVESTIGATION INTO PROBABILITY SPACE OF THE DIFFUSION MODELS

In Sec. 4.1, we argued that 1) since the user-specified condition is usually not observed in the training procedure of diffusion, an image x properly generated from a condition is actually out of distribution in the training distribution, resulting in a lower measured $p_{\boldsymbol{\theta}}(\mathbf{x})$, 2) this effect is so strong that it biases $p_{\boldsymbol{\theta}}(\mathbf{x}|\mathbf{c})$, and 3) by debiasing the conditional probability with the probability, i.e., $CAS = \log p_{\boldsymbol{\theta}}(\mathbf{x}|\mathbf{c}) - \log p_{\boldsymbol{\theta}}(\mathbf{x})$, it becomes possible to measure alignment between image and conditions. In this section, we provide a toy experiment to support these assertions. We selected two prompts, one representing an unlikely scenario in the real world and the other a more plausible scenario. The first prompt, $\mathbf{c}_1$, is "a woman with black hair," and the second, $\mathbf{c}_2$, is "a woman with rainbow hair." Then, we generated 100 images from each prompt and measured their average $\log p_{\boldsymbol{\theta}}(\mathbf{x}|\mathbf{c})$, $\log p_{\boldsymbol{\theta}}(\mathbf{x})$, and $CAS(\mathbf{x},\mathbf{c},\boldsymbol{\theta})$. As shown in Table 4, the results satisfy all aspects of our hypothesis: 1) $\mathbb{E}[\log p_{\boldsymbol{\theta}}(\mathbf{x}_1)] > \mathbb{E}[\log p_{\boldsymbol{\theta}}(\mathbf{x}_2)]$, indicating that less observed conditions result in a lower probability, 2) the similarity between $\mathbb{E}[\log p_{\boldsymbol{\theta}}(\mathbf{x})]$ and $\mathbb{E}[\log p_{\boldsymbol{\theta}}(\mathbf{x}|c)]$ suggests that $\log p_{\boldsymbol{\theta}}(\mathbf{x}|c)$ is biased by $\log p_{\boldsymbol{\theta}}(\mathbf{x})$, and 3) $\mathbb{E}[CAS(\mathbf{x}_1,\mathbf{c}_1,\boldsymbol{\theta})] > \mathbb{E}[CAS(\mathbf{x}_2,\mathbf{c}_1,\boldsymbol{\theta})]$ and $\mathbb{E}[CAS(\mathbf{x}_2,\mathbf{c}_2,\boldsymbol{\theta})] > \mathbb{E}[CAS(\mathbf{x}_1,\mathbf{c}_2,\boldsymbol{\theta})]$ demonstrate that CAS effectively measures the alignment between images and conditions.

### A.2  THE ABILITY OF CAS TO DETECT OUT-OF-DISTRIBUTION SAMPLES

In this section, we further investigate the capability of CAS to detect out-of-distribution samples and apply this to fake detection and image generation source detection. In Table 2 our technique significantly outperforms other methods in the Van Gogh dataset. This indicates the superior ability of our approach to identify samples that are out-of-distribution from the training distribution of the diffusion network. To further analyze this, we design the following experiment. We generated 200 images from each of the three different diffusion models, Dreamlike Photoreal 2.0 (DP2.0) (Art, 2023), Open Journey (OJ) (Prompthero, 2023), and Stable Diffusion 1.5 (SD1.5) (Rombach et al., 2022). We have also prepared 200 real images. Then, we measured the average CAS of each group of images according to the diffusion models for image generation, and CAS is computed with respective diffusion models that are used for generating each group of images. As shown in Table 5, CAS is significantly higher for images generated by the same diffusion model that CAS employs. Therefore, in the Van Gogh dataset, the scores are biased towards the Van Gogh images,

| Detection Type | Category | Image Source | | |
|---|---|---|---|---|
| | | Real | DP2.0/RV1.4/EPIC/SD2.1 | SD1.5/OJ |
| Source Detection | # of Train Samples | 100 | 100 per each model | None |
| | # of Test Samples | 200 | 200 per each model | None |
| Fake Detection | # of Train Samples | 100 | 100 per each model | None |
| | # of Test Samples | 200 | None | 200 per each model |

Table 6: Composition of train and test data used for source detection and fake detection. For dataset construction, we randomly sampled 300 (caption, image) pairs from the COCO validation dataset. For each caption, we generated images using six diffusion models: Dreamlike Photoreal 2.0 (DP2.0) (Art, 2023), Realistic Vision 1.4 (RV1.4) (SG1612222023, 2023), EPIC diffusion (EPIC) (Slegers, 2023), Open Journey (OJ) (Prompthero, 2023), Stable Diffusion 1.5 (SD1.5) (Rombach et al., 2022), and Stable Diffusion 2.1 (SD2.1) (Rombach et al., 2022), and then divided them into train and test sets.

| Method | Accuracy | Average Precision |
|---|---|---|
| CLIP-ViT (LC) | 0.87 | 0.91 |
| Ours | **0.92** | **0.96** |

Table 7: Result of fake image detection. CLIP-ViT(LC) (Ojha et al., 2023) is the current state-of-the-art in fake detection.

| Source | Real | DP2.0 | RV1.4 | EPIC | SD2.1 | Total |
|---|---|---|---|---|---|---|
| Accuracy | 0.87 | 0.86 | 0.90 | 0.92 | 0.95 | 0.90 |

Table 8: Result of image generation source detection.

leading to higher CAS accuracy. Furthermore, the characteristic of CAS producing high values for in-distribution samples and low values for real images shown in Table 5 can be straightforwardly applied to fake image detection and image generation source detection. Through a small-scale experiment, we show the potential of our method to surpass the current state-of-the-art (Ojha et al., 2023) in fake detection.

For fake detection and image generation source classification, we built a small custom dataset by randomly sampling (image, caption) pairs from the COCO validation dataset and generating images using various diffusion models and captions. The construction of this dataset is well described in Table. 6. The inference process is straightforward: we measure CAS from images using four diffusion models (DP2.0, RV1.4, EPIC and SD2.1) and feed these into a 3-Layer MLP model to generate the prediction. During training, the diffusion model is frozen, and only the MLP is trained. For a baseline in fake detection, we used the current state-of-the-art, CLIP-ViT (LC) (Ojha et al., 2023), trained on the same dataset. As shown in Table. 7, this CAS ensemble MLP model achieves an accuracy of 0.92 with only 500 training samples, surpassing the performance of CLIP-ViT (LC), which is 0.87. It is remarkable that the model performs well even though the diffusion models used for generating the test and training samples are different. Moreover, as shown in Table 8, the performance of image generation source classification also reaches an accuracy of 0.90.

The characteristic of CAS measuring lower for out-of-distribution samples in the diffusion model training distribution can be applied to fake detection models and image generation source classification models.

| Model $\backslash \sigma$ | 0.1 | 0.01 | 0.001 | 0.0001 | 0.00001 |
|---|---|---|---|---|---|
| SD2.1 | 1.240 | 0.237 | 0.033 | 0.020 | 0.070 |
| DP2.0 | 0.305 | 0.053 | 0.005 | 0.008 | 0.198 |

Table 9: Normalized Root Mean Square Error (NRMSE) between log probability computed using Song et al. (2021b) and log probability computed using our method, across the Dreamlike Photoreal 2.0 (DP2.0) (Art, 2023) and Stable Diffusion 2.1 (SD2.1) (Rombach et al., 2022).

| $\sigma$ | $10^{-2}$ | $10^{-3}$ | $10^{-4}$ | $10^{-5}$ | $10^{-6}$ | $10^{-7}$ |
|---|---|---|---|---|---|---|
| Ratio | 0.0 | 0.0 | $9.16 \cdot 10^{-7}$ | $9.16 \cdot 10^{-06}$ | $8.85 \cdot 10^{-05}$ | 0.0018 |

Table 10: The proportion of each element of the estimated gradient being zero, on various $\sigma$.

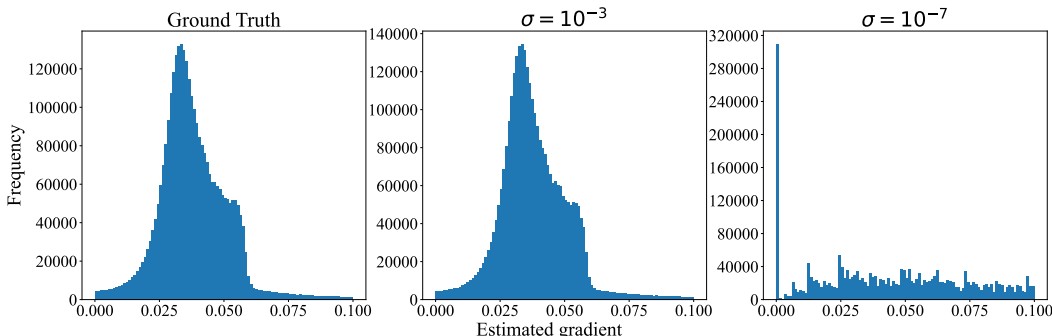

Figure 7: Histogram of elements in estimated gradient for varying $\sigma$.

### A.3 ANALYSIS ON OPTIMAL $\sigma$ FOR APPROXIMATION OF EQUATION 10

In Sec. 4.4, we provided the method to approximate $\mathbf{z}^\top \nabla_{\mathbf{x}} \boldsymbol{\epsilon}_{\boldsymbol{\theta}}(\mathbf{x}, t) \mathbf{z}$, avoiding the resource-intensive backpropagation process:

$$\mathbf{z}^\top \nabla_{\mathbf{x}} \boldsymbol{\epsilon}_{\boldsymbol{\theta}}(\mathbf{x}, t) \mathbf{z} \simeq \mathbf{z}^\top \frac{\boldsymbol{\epsilon}_{\boldsymbol{\theta}}(\mathbf{x} + \sigma \mathbf{z}, t) - \boldsymbol{\epsilon}_{\boldsymbol{\theta}}(\mathbf{x}, t)}{\sigma} = \frac{1}{\sigma} \mathbf{z}^\top (\boldsymbol{\epsilon}_{\boldsymbol{\theta}}(\mathbf{x} + \sigma \mathbf{z}, t) - \boldsymbol{\epsilon}_{\boldsymbol{\theta}}(\mathbf{x}, t)). \qquad (11)$$

We set $\sigma$ to $10^{-3}$ which shows the best approximation performance in Table 15. However, there are two ambiguities: the optimal $\sigma$ may vary for each diffusion model, and theoretically, a smaller $\sigma$ should yield better performance, which does not align with our experimental results. Therefore, in this section, we explore finding the optimal $\sigma$ and analyze the reasons behind its optimality. In our quest to identify the optimal $\sigma$ for model performance, we extended our ablation study to include two additional models, SD2.1 and DP2.0. Our findings, detailed in Table 9, reveal that a $\sigma$ value of 0.001 consistently yields the most accurate predictions of log probability across these models.

The core principle of differentiation suggests that as $\sigma$ approaches zero, the Normalized Root Mean Square Error (NRMSE) should correspondingly decrease. Contrary to this expectation, we observed an increase in error when $\sigma$ was set to $10^{-7}$. This anomaly led us to investigate the underlying causes. This discrepancy can be attributed to the precision limitations of the float32 format. At very low $\sigma$ values, subtraction operations lead to significant rounding errors, nullifying the computed gradient. This phenomenon became apparent during our analysis of gradient calculations, where we observed a noticeable increase in zero-valued gradient elements as $\sigma$ approached zero. Our further analysis includes a comparison of the distribution of elements within the estimated gradient vector. We plot the histogram of elements in the estimated gradient, particularly focusing on the proportion of elements that are zero in different $\sigma$ scenarios. Fig. 7 and Table 10 illustrate these findings, showing a distinct rise in the frequency of elements near zero. Notably, while the distribution of estimated gradients at $\sigma = 10^{-3}$ is relatively similar to that of the ground truth, the distribution of estimated gradients at $\sigma = 10^{-7}$ is quite different from that of the ground truth. The estimated gradients are concentrated near zero.

| | CLIP Score | Image Reward | HPS | Pick Score | Ours |
|---|---|---|---|---|---|
| EER on Pick Score dataset | 0.419 | 0.383 | 0.306 | **0.278** | 0.376 |

Table 11: EER for each metric on our conducted Pick Score dataset.

| Model | InstructPix2Pix | Canny Edge ControlNet | Scribble ControlNet | AudioLDM |
|---|---|---|---|---|
| p-value | $1.74 \cdot 10^{-8}$ | $3.18 \cdot 10^{-18}$ | $2.16 \cdot 10^{-11}$ | $3.22 \cdot 10^{-7}$ |

Table 12: Statistical significance of human preference evaluation result for each modalities.

| | CAS score percentile | | | |
|---|---|---|---|---|
| Model | 0~25% (Lowest) | 25~50% | 50~75% | 75~100% (Highest) |
| InstructPix2Pix | 0.341 | 0.489 | 0.568 | 0.6 |
| ControlNet (Canny Edge) | 0.4875 | 0.4875 | 0.5 | 0.525 |
| ControlNet (Scribble) | 0.425 | 0.45 | 0.55 | 0.575 |
| AudioLDM | 0.4 | 0.439 | 0.575 | 0.585 |

Table 13: Histogram of average human scores divided based on CAS score percentiles.

This issue is a recognized challenge in numerical differentiation and stems from the inherent precision constraints of float32. Consequently, adopting a $\sigma$ value of $10^{-3}$ is a widely accepted compromise in addressing these computational limitations. Therefore, we can conclude that our setting of $\sigma = 10^{-3}$ is the promising selection.

## B ADDITIONAL EXPERIMENTS

### B.1 EER METRIC ON PICKSCORE DATASET

We evaluated the Equal Error Rate (EER) of various methods, including our proposed approach, using the Pickscore dataset. Since the task in the Pick Score dataset is slightly different from typical binary classification, we measured EER albeit with a slight modification. For each image pair (image 1, image 2), the preferred image was assigned a score of 1, and the less preferred one a score of 0. We then calculated the prediction scores as the difference between the scores of the two images (*i.e.*, score(image 1) - score(image 2) for image 1, and vice versa for image 2). Using these predictions and the designated human preference scores, we computed the EER for each method, providing a nuanced comparison of their performance. As shown in Table 11, EER measured by this scheme follows the behavior of accuracy.

### B.2 ADDITIONAL RESULTS OF OTHER MODALITIES

**Statistical Significance of Model Performance** We conducted a binomial test on the human evaluation results for a multimodal dataset to assess the probability of the models performing significantly better than random guessing, assumed to be 50% accuracy. The p-value, which measures the probability of obtaining a result at least as extreme as the one observed under the assumption of random guessing, was calculated for each modalities. As shown in Table 12, the p-values for all modalities are below 0.05. This indicates that the performance on each modalities is statistically significantly better than random chance, demonstrating the effectiveness of our method across various modalities.

**Correlation Between CAS and Human Preferences** In Sec. 5.2, we measured the performance of CAS across various modalities by measuring alignments with the preference of five participants for each test case. Based on these preferences, we analyze the correlation between CAS and mean human preference. Dividing the CAS scores into histogram intervals based on percentiles, the corre-

| $\lambda$ | 0.8 | 0.9 | 1.0 | 1.1 |
|---|---|---|---|---|
| Accuracy | 0.532 | 0.525 | **0.622** | 0.494 |

Table 14: Accuracy of our method on Pickscore dataset, varying $\lambda$.

| | DDIM Inversion | | | 2nd Order Recursive Inversion | | |
|---|---|---|---|---|---|---|
| Step | 10 | 50 | 100 | 10 | 50 | 100 |
| Accuracy | 0.593 | 0.589 | 0.585 | **0.622** | 0.601 | 0.591 |
| Time (s) | 31.3 | 171.4 | 347.2 | 33.2 | 180.2 | 364.5 |

Table 15: Accuracy in Pick Score dataset and average computation time spent for one sample with various settings of the inversion process.

sponding average human score is measured. As shown in Table 13, there is a tendency for the mean human score to increase with higher CAS scores, indicating a correlation between CAS and human preference.

## C   ABLATION STUDY

### C.1   PERFORMANCE OF CAS WITH CHOICE OF DIFFERENT $\lambda$

We conducted an ablation study of $\lambda$ term on $CAS(\mathbf{x}, \mathbf{c}, \theta) = \log p_{\theta}(\mathbf{x}|\mathbf{c}) - \lambda \log p_{\theta}(\mathbf{x})$. Following the experiment setting of Sec.5.1, we computed accuracy of our method on Pick Score dataset when $\lambda = 0.8,\ 0.9,\ 1.0,$ and $1.1$. As shown in Table 14, the accuracy is highest when $\lambda = 1.0$.

### C.2   PERFORMANCE OF CAS USING VARIOUS INVERSION PROCESS

The accuracy of CAS in Pick Score dataset provided in Sec. 5.1 is based on the 2nd order recursive inversion with a total step of 10. In this section, we provide the performance of CAS derived from inversions with different hyperparameters.

The results are presented in Table 15. From the standpoint of computation time, utilizing the 2nd order inversion is considerably more cheaper than simply increasing the steps in DDIM inversion. This is predominantly due to the fact that the time required to compute $\mathbb{E}_{p(\mathbf{z})}[\mathbf{z}^{\top}\nabla_{\mathbf{x}}\epsilon_{\theta}(\mathbf{x}, t)\mathbf{z}]$ vastly outweighs the inversion process itself. We can roughly infer the computation time based on the number of times the forward function of the diffusion model $\theta$ is called for computing $\epsilon_{\theta}$. If we denote the number of steps as $n$, for an $i$-th order recursive inversion process, the forward function is called $ni$ times. Meanwhile, to compute $\mathbb{E}_{p(\mathbf{z})}[\mathbf{z}^{\top}\nabla_{\mathbf{x}}\epsilon_{\theta}(\mathbf{x}, t)\mathbf{z}]$, the forward function is called $nk$ times where $k$ is number of sampling of random variable $\mathbf{z}$. This results in a total of $n(i + k)$ calls. However, we find that setting $k$ to 20 is optimal for stable experimental outcomes. Therefore, it is evident that increasing $n$ incurs a significantly greater computational overhead than increasing $k$, *i.e.* recursive inversion is cheaper than simply increasing the number of inversion steps. This aligns with the results presented in Table 15.

From an accuracy standpoint, we observe the following: 1) The 2nd order recursive inversion consistently outperforms the DDIM inversion, and 2) as the step size increases, accuracy diminishes. The superior accuracy of the 2nd order recursive inversion signifies that our methodology is functioning as anticipated. The decline in accuracy with an increased step size in DDIM inversion suggests that merely enlarging the step size is insufficient for enhancing accuracy. In contrast, our approach manifests a boost in accuracy.

## D THE DETAILED DERIVATIONS

### D.1 THE FULL DERIVATION OF CAS

In this section, we provide the detailed derivation of CAS in order. Please refer to Sec. 3 and Sec. 4 for the definition of symbols. Before deriving CAS, to recap, DDIM (Song et al., 2021a) is defined as follows:

$$\mathbf{x}_{t-1} = \sqrt{\alpha_{t-1}}(\frac{\mathbf{x}_t - \sqrt{1 - \alpha_t}\boldsymbol{\epsilon_\theta}(\mathbf{x}_t, t)}{\sqrt{\alpha_t}}) + \sqrt{1 - \alpha_{t-1}}\boldsymbol{\epsilon_\theta}(\mathbf{x}_t, t). \tag{12}$$

where $t = 1, 2, ..., N$ and $\sigma_t$ is set as 0 for all $t$ in equation 1. To recap, the probability of generated sample $\mathbf{x}(0)$ can be calculated by:

$$\log p_0(\mathbf{x}(0)) = \log p_1(\mathbf{x}(1)) + \int_0^1 \nabla_{\mathbf{x}} \cdot \mathbf{f_\theta}(\mathbf{x}(t), t)dt, \tag{13}$$

where probability flow ODE is given as follows: $d\mathbf{x} = \mathbf{f_\theta}(\mathbf{x}(t), t)dt$ (Song et al., 2021b). The time $t$ is a continuous variable whose interval is defined as $[0, 1]$ here.

The framework for calculating CAS is well illustrated in Fig. 3, which involves: 1) undergoing DDIM inversion for the image $\mathbf{x}$ and storing $\{\boldsymbol{\epsilon_\theta}(\mathbf{x}_t, \mathbf{c}, t)\}_{t=1}^N$, $\{\boldsymbol{\epsilon_\theta}(\mathbf{x}_t, \emptyset, t)\}_{t=1}^N$, $\mathbf{x}_N^c$ and $\mathbf{x}_N^u$, 2) applying equation 13 based on these values to compute $\log p_\theta(\mathbf{x})$ and $\log p_\theta(\mathbf{x}|\mathbf{c})$, and 3) calculating $CAS(\mathbf{x}, \mathbf{c}, \boldsymbol{\theta}) = \log p_\theta(\mathbf{x}|\mathbf{c}) - \log p_\theta(\mathbf{x})$. As mentioned above, equation 13 is calculated under the ODE of the diffusion process, which treats time as continuous, however, DDIM treats time as discrete. Therefore, DDIM should be rewritten in the form of ODE. As Liu et al. (2022) did, we replace discrete $t - 1$ with a continuous version $t - \delta$ and subtract $\mathbf{x}_t$ from both sides of the equation to get the differential form:

$$\mathbf{x}_{t-\delta} - \mathbf{x}_t = (\alpha_{t-\delta} - \alpha_t)(\frac{\mathbf{x}_t}{\sqrt{\alpha_t}(\sqrt{\alpha_{t-\delta}} + \sqrt{\alpha_t})} - \frac{\boldsymbol{\epsilon_\theta}(\mathbf{x}_t, t)}{\sqrt{\alpha_t}(\sqrt{(1 - \alpha_{t-\delta})\alpha_t} + \sqrt{(1 - \alpha_t)\alpha_{t-\delta}})}). \tag{14}$$

Then, dividing equation 14 by $\delta$ and taking $\delta$ to zero, we can get the corresponding ODE:

$$\frac{d\mathbf{x}}{dt} = \lim_{\delta \to 0} \frac{\mathbf{x}_t - \mathbf{x}_{t-\delta}}{\delta} = \alpha'(t) \left( \frac{\mathbf{x}(t)}{2\alpha(t)} - \frac{\boldsymbol{\epsilon_\theta}(\mathbf{x}(t), t)}{2\alpha(t)\sqrt{1 - \alpha(t)}} \right) \tag{15}$$

where $\mathbf{x}(t)$ and $\alpha(t)$ are the continuous versions of $\{\mathbf{x}_t\}_{t=1}^N$ and $\{\alpha_t\}_{t=1}^N$ with time interval defined as $[0,1]$. Applying equation 15 to equation 13, we can compute the log probability in DDIM inversion process:

$$\log p_0(\mathbf{x}(0)) = \log p_1(\mathbf{x}(1)) + \int_0^1 \nabla_{\mathbf{x}} \cdot \left( \alpha'(t) \left( \frac{\mathbf{x}(t)}{2\alpha(t)} - \frac{\boldsymbol{\epsilon_\theta}(\mathbf{x}(t), t)}{2\alpha(t)\sqrt{1 - \alpha(t)}} \right) \right) dt \tag{16}$$

$$= \log p_1(\mathbf{x}(1)) + \int_0^1 \alpha'(t) \left( \frac{\nabla_{\mathbf{x}} \cdot \mathbf{x}(t)}{2\alpha(t)} \right) dt - \int_0^1 \alpha'(t) \frac{\nabla_{\mathbf{x}} \cdot \boldsymbol{\epsilon_\theta}(\mathbf{x}(t), t)}{2\alpha(t)\sqrt{1 - \alpha(t)}} dt \tag{17}$$

$$= \log p_1(\mathbf{x}(1)) + \int_{\alpha(0)}^{\alpha(1)} \frac{D}{2\alpha} d\alpha - \int_0^1 \alpha'(t) \frac{\nabla_{\mathbf{x}} \cdot \boldsymbol{\epsilon_\theta}(\mathbf{x}(t), t)}{2\alpha(t)\sqrt{1 - \alpha(t)}} dt \tag{18}$$

$$= \log p_1(\mathbf{x}(1)) + \log \frac{\sqrt{\alpha(0)}}{\sqrt{\alpha(1)}} D - \int_0^1 \alpha'(t) \frac{\nabla_{\mathbf{x}} \cdot \boldsymbol{\epsilon_\theta}(\mathbf{x}(t), t)}{2\alpha(t)\sqrt{1 - \alpha(t)}} dt, \tag{19}$$

where $D$ is the dimension of image $\mathbf{x}(0)$. By applying this formula, we can derive both $\log p_\theta(\mathbf{x})$ and $\log p_\theta(\mathbf{x}|c)$ from the values obtained through DDIM inversion, $\{\boldsymbol{\epsilon_\theta}(\mathbf{x}_t, t, \mathbf{c})\}_{t=1}^N$, $\{\boldsymbol{\epsilon_\theta}(\mathbf{x}_t, t, \emptyset)\}_{t=1}^N$, $\mathbf{x}_N^c$ and $\mathbf{x}_N^u$. When $N$ is large, by transferring the time domain from $[1, N]$ to $[0, 1]$, we can convert $\mathbf{x}_t^u$ and $\mathbf{x}_t^c$ into continuous variables with respect to t as $\mathbf{x}^u(t/N)$ and $\mathbf{x}^c(t/N)$. Then, $p_0(\mathbf{x}^c(0)|c)$ and $p_0(\mathbf{x}^u(0))$ are derived as:

$$\log p_0(\mathbf{x}^u(0)) = \log p_1(\mathbf{x}^u(1)) + \log \frac{\sqrt{\alpha(0)}}{\sqrt{\alpha(1)}} D - \int_0^1 \alpha'(t) \frac{\nabla_x \cdot \boldsymbol{\epsilon_\theta}(\mathbf{x}^u(t), \emptyset, t)}{2\alpha(t)\sqrt{1 - \alpha(t)}} dt, \tag{20}$$

$$\log p_0(\mathbf{x}^c(0)|\mathbf{c}) = \log p_1(\mathbf{x}^c(1)|\mathbf{c}) + \log \frac{\sqrt{\alpha(0)}}{\sqrt{\alpha(1)}} D - \int_0^1 \alpha'(t) \frac{\nabla_{\mathbf{x}} \cdot \boldsymbol{\epsilon_\theta}(\mathbf{x}^c(t), \mathbf{c}, t)}{2\alpha(t)\sqrt{1 - \alpha(t)}} dt. \tag{21}$$

Subtracting equation 20 from equation 21, CAS can be derived as:

$$CAS(\mathbf{x}, \mathbf{c}, \boldsymbol{\theta}) = \log \frac{p_1(\mathbf{x}^c(1)|\mathbf{c})}{p_1(\mathbf{x}^u(1))} - \int_0^1 \alpha'(t) \frac{\nabla_\mathbf{x} \cdot \boldsymbol{\epsilon_\theta}(\mathbf{x}^c(t), \mathbf{c}, t) - \nabla_\mathbf{x} \cdot \boldsymbol{\epsilon_\theta}(\mathbf{x}^u(t), \emptyset, t)}{2\alpha(t)\sqrt{1 - \alpha(t)}} dt, \quad (22)$$

where $\nabla_\mathbf{x} \cdot \boldsymbol{\epsilon_\theta}(\mathbf{x}^c(t), \mathbf{c}, t)$ and $\nabla_\mathbf{x} \cdot \boldsymbol{\epsilon_\theta}(\mathbf{x}^u(t), \emptyset, t)$ can be computed by Skilling-Hutchinson trace estimator (Skilling, 1989; Hutchinson, 1989), $\nabla_\mathbf{x} \cdot \boldsymbol{\epsilon_\theta}(\mathbf{x}(t), t) = \mathbb{E}_{p(\mathbf{z})}[\mathbf{z}^\top \nabla_\mathbf{x} \boldsymbol{\epsilon_\theta}(\mathbf{x}, t)\mathbf{z}]$ with respect to our approximation method introduced in equation 10. With the obtained values from the Skilling-Hutchinson estimator, the numerical integration method is applied to obtain CAS.

## D.2 DETAILED EXPLANATION OF DDIM RECURSIVE INVERSION

In this section, we provide the underlying idea of the DDIM recursive inversion. To recap, the ideal DDIM inversion is given as follows:

$$\mathbf{x}_t = a_{t-1}\mathbf{x}_{t-1} + b_{t-1}\boldsymbol{\epsilon_\theta}(\mathbf{x}_t, t), \quad (23)$$

where $a_{t-1} = \frac{\sqrt{\alpha_t}}{\sqrt{\alpha_{t-1}}}$, $b_{t-1} = (\sqrt{1 - \alpha_t} - \sqrt{\frac{\alpha_t}{\alpha_{t-1}} - \alpha_t})$, and $\sigma_t = 0$. However, it is contradictory to use $x_t$ itself to recover $x_t$. Therefore, it is convention to approximate it by:

$$\hat{\mathbf{x}}_t = a_{t-1}\mathbf{x}_{t-1} + b_{t-1}\boldsymbol{\epsilon_\theta}(\mathbf{x}_{t-1}, t), \quad (24)$$

Our DDIM recursive inversion is defined as follows:

$$\hat{\mathbf{x}}_t^i = a_{t-1}\mathbf{x}_{t-1} + b_{t-1}\boldsymbol{\epsilon_\theta}(\hat{\mathbf{x}}_t^{i-1}, t) \quad (25)$$

for $i = 2, 3, ...$ and $\hat{\mathbf{x}}_t^1 = \hat{\mathbf{x}}_t$, and we denote the approximation $\mathbf{x}_t \approx \hat{\mathbf{x}}_t^n$ as the $n$-th order recursive inversion.

To effectively illustrate the underlying idea of this DDIM resursive inversion, we consider the simple case that: 1) $\hat{\mathbf{x}}_t$, the latent derived by DDIM inversion is closer to true latent $\mathbf{x}_t$ than $\mathbf{x}_{t-1}$, *i.e.* $||\mathbf{x}_t - \hat{\mathbf{x}}_t|| < ||\mathbf{x}_t - \mathbf{x}_{t-1}||$, and 2) for two random vectors $\mathbf{u}$ and $\mathbf{v}$ close to $\mathbf{x}_t$, if $||\mathbf{x}_t - \mathbf{u}|| < ||\mathbf{x}_t - \mathbf{v}||$, then $||\boldsymbol{\epsilon_\theta}(\mathbf{x}_t, t) - \boldsymbol{\epsilon_\theta}(\mathbf{u}, t)|| < ||\boldsymbol{\epsilon_\theta}(\mathbf{x}_t, t) - \boldsymbol{\epsilon_\theta}(\mathbf{v}, t)||$ holds. Subtracting the ideal DDIM inversion (equation 23) from 2nd order DDIM inversion ($\hat{\mathbf{x}}_t^2$ in equation 25), the following equations can be derived:

$$\begin{aligned}
||\hat{\mathbf{x}}_t^2 - \mathbf{x}_t|| &= b_{t-1}||\boldsymbol{\epsilon_\theta}(\hat{\mathbf{x}}_t, t) - \boldsymbol{\epsilon_\theta}(\mathbf{x}_t, t)|| \\
&< b_{t-1}||\boldsymbol{\epsilon_\theta}(\mathbf{x}_{t-1}, t) - \boldsymbol{\epsilon_\theta}(\mathbf{x}_t, t)|| \\
&= ||\hat{\mathbf{x}}_t - \mathbf{x}_t|| 
\end{aligned} \quad (26)$$

by the assumptions. Assuming that $||\hat{\mathbf{x}}_t^{i-1} - \mathbf{x}|| < ||\hat{\mathbf{x}}_t^{i-2} - \mathbf{x}||$, the following equations can be derived:

$$\begin{aligned}
||\hat{\mathbf{x}}_t^i - \mathbf{x}_t|| &= b_{t-1}||\boldsymbol{\epsilon_\theta}(\hat{\mathbf{x}}_t^{i-1}, t) - \boldsymbol{\epsilon_\theta}(\mathbf{x}_t, t)|| \\
&< b_{t-1}||\boldsymbol{\epsilon_\theta}(\mathbf{x}_t^{i-2}, t) - \boldsymbol{\epsilon_\theta}(\mathbf{x}_t, t)|| \\
&= ||\hat{\mathbf{x}}_t^{i-1} - \mathbf{x}_t||. 
\end{aligned} \quad (27)$$

By applying equation 27 recursively on equation 26, we can expect that the error between the true latent and predicted latent decreases, *i.e.*, $||\hat{\mathbf{x}}_t - \mathbf{x}_t|| > ||\hat{\mathbf{x}}_t^2 - \mathbf{x}_t|| > ||\hat{\mathbf{x}}_t^3 - \mathbf{x}_t|| > ...$ holds. Note that we use multiple assumptions for simplicity, which effectively illustrates the underlying idea of the DDIM recursive inversion.

## E RANKING EXAMPLES OF CAS

We have prepared several examples demonstrating the ranking of CAS across various modalities, including text-to-image, image-and-instruction to image, edge-and-prompt to image, and scribble-and-prompt to image.

Condition High rank Ranked Images

"A winter wonderland at night, with ice sculptures glowing under the **aurora borealis**, people skating on a frozen lake, and cozy igloos serving warm, spiced drinks."

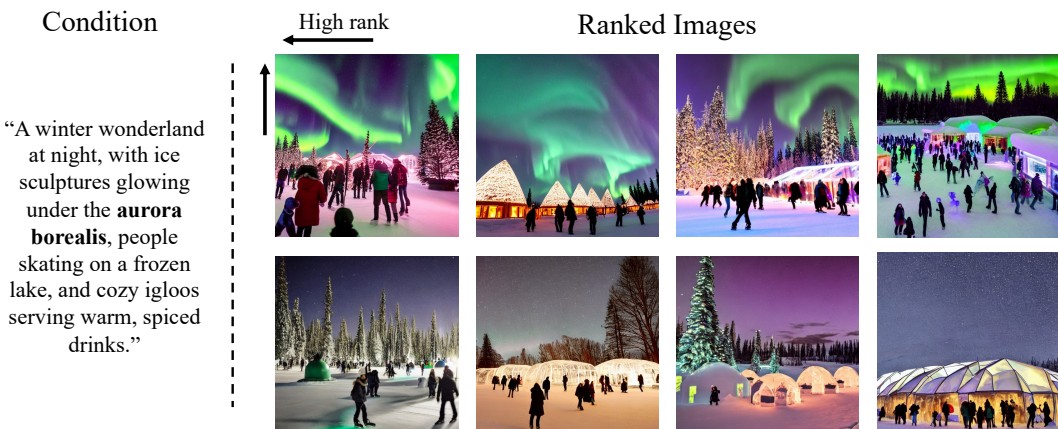

Figure 8: Text-to-image. Stable Diffusion v1.5(Rombach et al., 2022) is leveraged.

Condition High rank Ranked Images

"A serene beach at sunrise, with gentle waves, a clear sky, and a lone figure practicing yoga on the sand, with distant boats on the horizon."

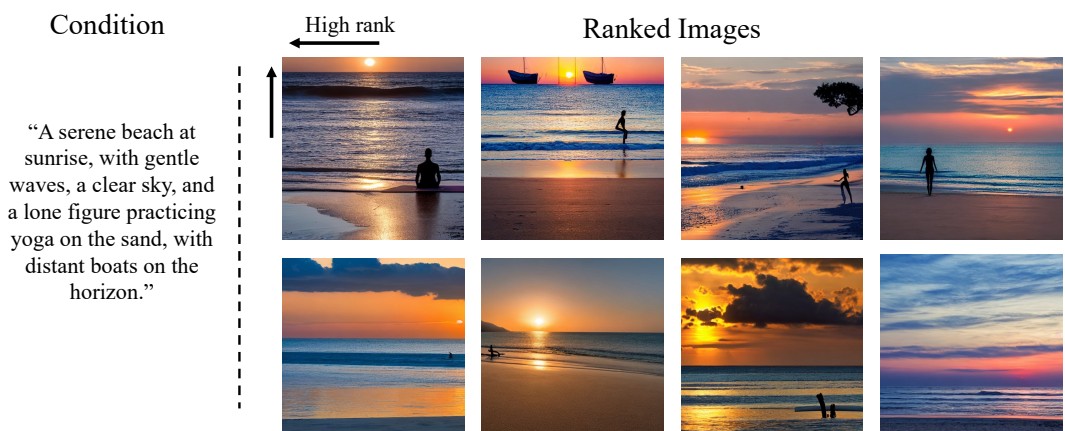

Figure 9: Text-to-image. Stable Diffusion v1.5(Rombach et al., 2022) is leveraged.

Condition High rank Ranked Images

"Make it a space suit"

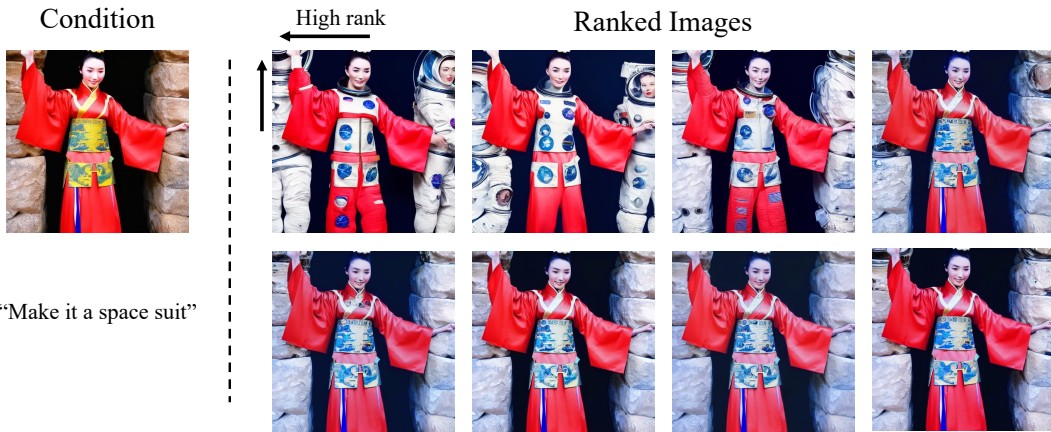

Figure 10: {Image, instruction}-to-image. InstructPix2xPix (Brooks et al., 2023) is leveraged.

Condition High rank Ranked Images

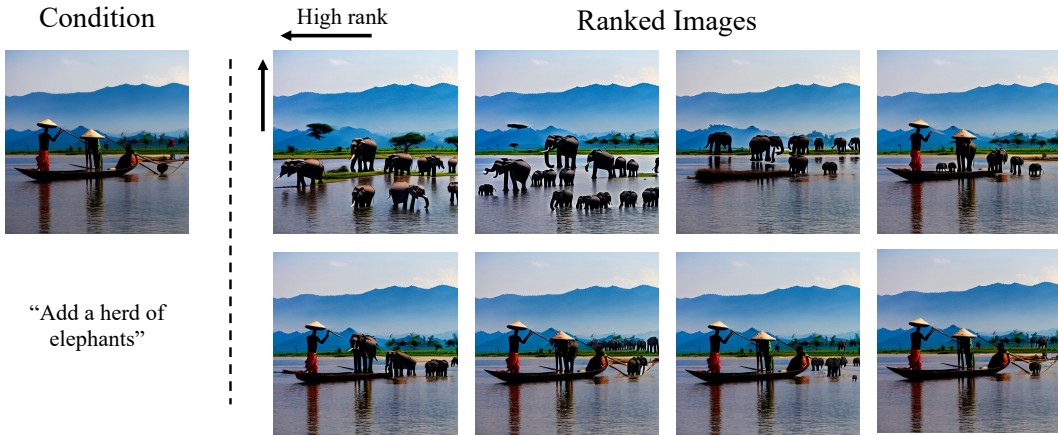

"Add a herd of elephants"

Figure 11: {Image, instruction}-to-image. InstructPix2xPix (Brooks et al., 2023) is leveraged.

Condition High rank Ranked Images

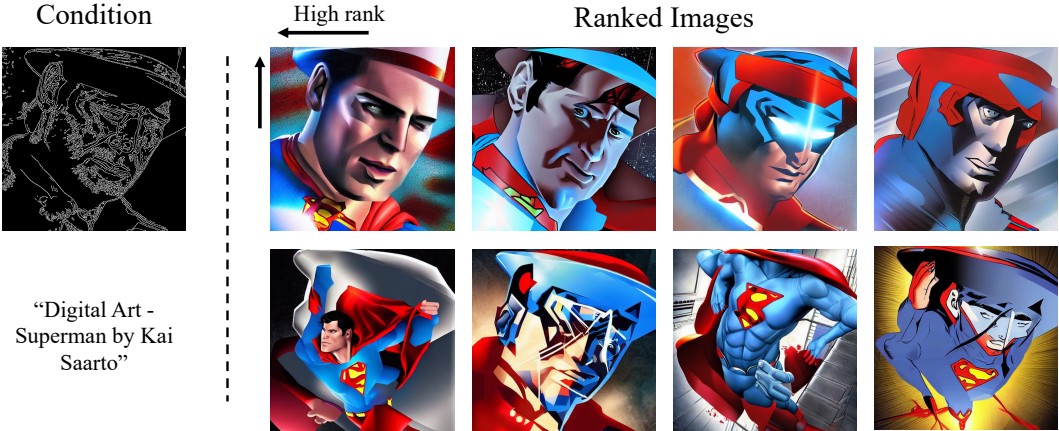

"Digital Art - Superman by Kai Saarto"

Figure 12: {edge, prompt}-to-image. Controlnet (Canny Edge) (Zhang et al., 2023) is leveraged.

Condition High rank Ranked Images

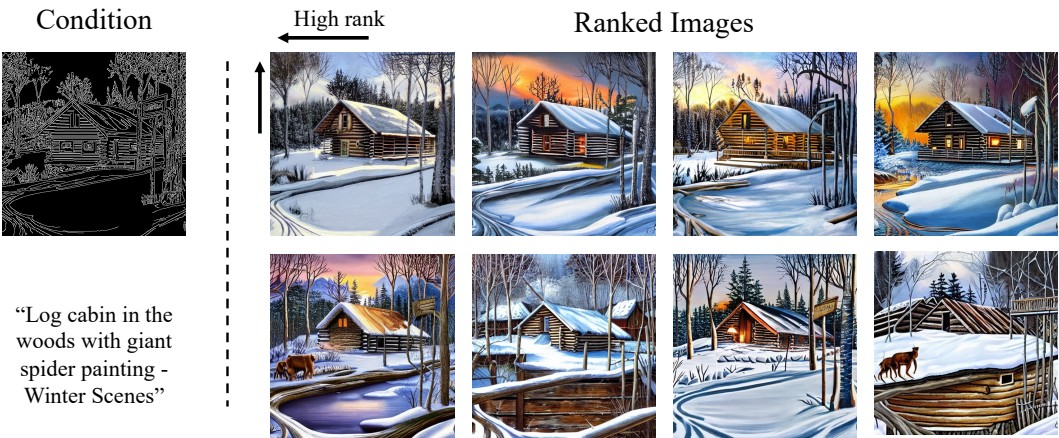

"Log cabin in the woods with giant spider painting - Winter Scenes"

Figure 13: {edge, prompt}-to-image. Controlnet (Canny Edge) (Zhang et al., 2023) is leveraged.

Condition

High rank

Ranked Images

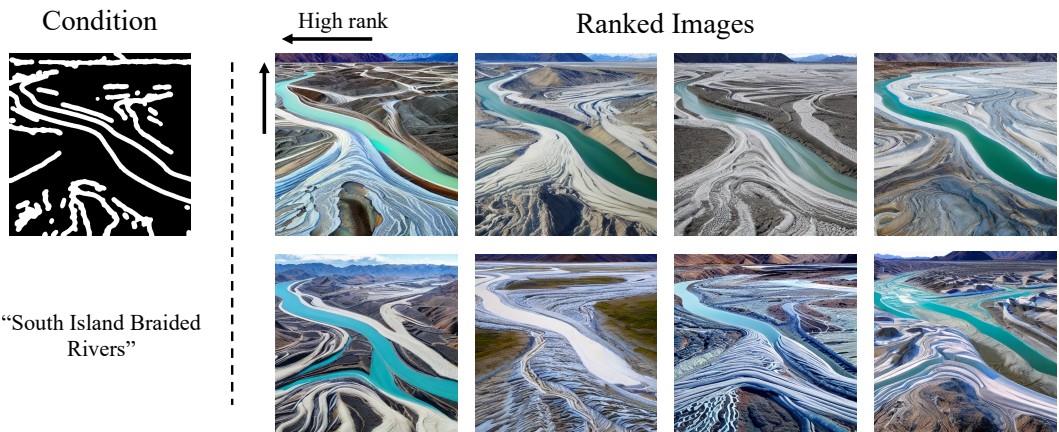

"South Island Braided Rivers"

Figure 14: {scribble, prompt}-to-image. Controlnet (Scribble) (Zhang et al., 2023) is leveraged.

Condition

High rank

Ranked Images

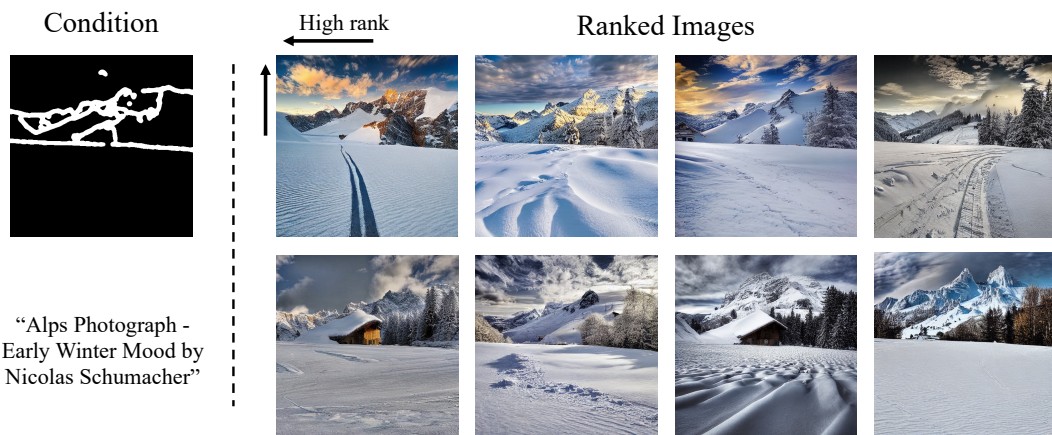

"Alps Photograph - Early Winter Mood by Nicolas Schumacher"

Figure 15: {scribble, prompt}-to-image. Controlnet (Scribble) (Zhang et al., 2023) is leveraged.

