# OpenReview forum: "CAS: A Probability-Based Approach for Universal Condition Alignment Score"
_ICLR.cc/2024/Conference — ICLR 2024 spotlight_

### Official Review · Reviewer_cbcH · 2023-10-30

**Soundness:** 4 excellent
**Presentation:** 3 good
**Contribution:** 3 good
**Rating:** 8
**Confidence:** 3

**Summary:**

This paper proposes a simple metric for measuring alignment between the generated sample and the condition for conditional diffusion models (such as text-to-image or text-to-audio models). The authors propose to leverage the conditional likelihood of the model which can be computed through the diffusion process, and name this CAS — condition alignment score. They also propose a method to improve the DDIM inversion that is required to compute the likelihood, and suggest a simple approximation to speed up this computation. CAS is measured for several domains, and found to align well with objective metrics like CLIP score or subjective human evaluations.

**Strengths:**

1. The problem of defining a universal metric for condition alignment is important but under-researched. Using the likelihood computed from the model itself is a clever choice, and is analogous to intrinsic measures like perplexity in language models.

2. The paper is well written and easy to understand, and the motivation for using the metric is explained clearly. The contributions for improving and speeding up DDIM inversion are also described well.

3. Evaluations are conducted across several domains including image and audio, and CAS shows strong correlation with previously used measures like CLIP score. This is useful for researchers using diffusion models in other domains.

4. Most importantly, the authors have mentioned that the code for computing CAS will be made publicly available. This should make it easy to use as a metric for future work on conditional diffusion models.

**Weaknesses:**

1. In Section 2 (paragraph 2), the authors mention that CAS score relies on computing the likelihood of samples generated from a conditional diffusion model, for which they use a method proposed previously by Song et al. This was later used by Zimmermann et al. to measure the class-conditional likelihood of images. The key difference, as described in Section 4.1, is that the authors subtract the unconditional likelihood. While the authors provide some empirical justification for this (Figure 2), it may be useful to conduct a more thorough investigation to justify the hypotheses proposed in Section 4.1, for example by training conditional diffusion models on some toy data.

2. In Section 4.4, the approximation only holds when $\sigma$ approaches 0. Based on this, we should expect the accuracy for the approximation to improve as we choose $\sigma$ closer to 0, but this is not true in Table 1. Do the authors have some explanation for this behavior? Further, given the large range of NRMSE values, how should practitioners choose this parameter when computing CAS for a new domain?

3. In Section 5.2, the authors only use a single metric (accuracy) to measure the correspondence between CAS and human preference. However, this does not give any insights on how well CAS difference correlates with human judgment. It may be more useful to show a plot with CAS difference between the samples and the corresponding human preference accuracy (for e.g., through a histogram), where we expect the accuracy to increase as the CAS difference gets larger.

**Questions:**

Will the dataset of human evaluations (used in Section 5.2) released along with the code?

---

> ### Author Response · Authors · 2023-11-20
> **Response to Reviewer cbcH (Part 1/2)**
>
> We thank the reviewer for the interest in our work and the valuable feedback that strengthens our paper. The segments of the revised paper that reflect the received feedback are marked in blue color.
>
> **W1. More investigation into the hypothesis which provided in Sec. 4.1. with toy experiments. (Appendix A.1)**
>
> As requested, we have newly conducted an additional toy experiment to support our claim in Sec. 4.1. Our objective is to demonstrate that 1) since the user-specified condition is usually not observed in the training procedure of diffusion, a generated image $\mathbf{x}$ complying with the given condition is actually often out-of-distribution in the training distribution, resulting in a lower measured $p_\mathbf{\theta}(\mathbf{x})$ , 2) this effect is so strong that it biases $p_\mathbf{\theta}(\mathbf{x}\vert\mathbf{c})$ , and 3) by debiasing the conditional probability with the probability, i.e., $CAS(\mathbf{x},\mathbf{c},\theta)=\log p_\mathbf{\theta}(\mathbf{x}\vert\mathbf{c}) - \log p_\mathbf{\theta}(\mathbf{x})$ , it facilitates to measure alignment between image and conditions.
>
> |    | Avg.  $\log p_\theta(\mathbf{x} \vert \mathbf{c}_1)$| Avg. $\log p_\theta(\mathbf{x} \vert \mathbf{c}_2)$| Avg. $\log p_\theta(\mathbf{x})$ | Avg. $CAS(\mathbf{x},\mathbf{c}_1)$ | Avg. $CAS(\mathbf{x},\mathbf{c}_2)$ |
> |----|-------------------|-------------------|---------------|----------------|----------------|
> | $\mathbf{x}_1$ | 61059.39          | 61046.07          | 61011.86      | 47.53          | 34.21          |
> | $\mathbf{x}_2$ | 59566.22          | 59612.24          | 59546.12      | 20.10          | 66.12          |
>
> We have conducted the following experiment: We generated 100 images each from two prompts, $\mathbf{c}_1=$ "a woman with black hair" and $\mathbf{c}_2=$"a woman with rainbow hair," denoted as $\mathbf{x}_1$ and $\mathbf{x}_2$, respectively.  $\mathbf{c}_2$ represents an unlikely scenario in the real world and $\mathbf{c}_1$ represents a more plausible scenario.
>
> We then measured the average $\log p_\mathbf{\theta}(\mathbf{x}|\mathbf{c})$, $\log p_\mathbf{\theta}(\mathbf{x})$, and $CAS(\mathbf{x},\mathbf{c},\theta)$ for these images.
>
> The results, as shown in the table above, satisfy all aspects of our hypothesis: 1) $\mathbb{E}[\log p_\theta(x_1)] > \mathbb{E}[\log p_\theta(x_2)]$, indicating that less observed conditions result in a lower probability, 2) the similarity between $\mathbb{E}[\log p_\mathbf{\theta}(\mathbf{x})]$ and $\mathbb{E}[\log p_\mathbf{\theta}(\mathbf{x}|\mathbf{c})]$ suggests that $\log p_\mathbf{\theta}(\mathbf{x}|\mathbf{c})$ is biased by $\log p_\mathbf{\theta}(\mathbf{x})$, and 3) $\mathbb{E}[CAS(\mathbf{x_1},\mathbf{c_1},\theta)]>\mathbb{E}[CAS(\mathbf{x_2},\mathbf{c_1},\theta)]$ and $\mathbb{E}[CAS(\mathbf{x_2},\mathbf{c_2},\theta)]>\mathbb{E}[CAS(\mathbf{x_1},\mathbf{c_2},\theta)]$ demonstrate that CAS effectively measures image condition alignment. We have newly added this experiment to Appendix A.1.
>
> **W2. Why NRMSE at the edge is big? How practitioners will select $\sigma$? (Appendix A.3)**
>
> The $\sigma$ is a hyper-parameter that controls the approximation quality of gradient by numerical differentiation. According to the definition of the numerical differentiation, it is obvious for Normalized Root Mean Square Error (NRMSE) to be large when $\sigma=10^{-1}$.
>
> However, the reason for the high value of NRMSE when $\sigma=10^{-7}$ is different, because the NRMSE should decrease, as $\sigma$ approaches zero. It has been well-known that with the limited precision number like float32 we used, when $\sigma$ is set to be too low, the numerical differentiation causes significant rounding errors, leading to the gradient being calculated as zero [C1]. Thus, the widely accepted and common practice is $\sigma=10^{-3}$ (refer to [C1]), which is consistent with our numerical analysis. This may hint that the sigma is a hyper-parameter that does not require intensive tuning at all.
>
> Furthermore, the table below shows the NRMSE between log probability computed by backpropagation and our approximation with various $\sigma$ for the Dreamlike Photoreal 2.0 model (Stable Diffusion v1.5 was employed in the NRMSE table in the main paper.) It can be observed that the approximation reaches the best performance when $\sigma=10^{-3}$. The results also support that practitioners can set $\sigma=10^{-3}$ regardless of the model. We have newly included these experiments in Appendix A.3 with additional details. Please note that this segment in the revised paper is colored green since there was the same question asked by Reveiwer uTB1.
>
> | $\sigma$ | $10^{-1}$  | $10^{-2}$  | $10^{-3}$  | $10^{-4}$  | $10^{-5}$  | $10^{-6}$  | $10^{-7}$  |
> |--------|--------|--------|--------|--------|--------|--------|--------|
> | NRMSE  | 0.3048 | 0.0529 | 0.0051 | 0.0081 | 0.1983 | 0.6777 | 8.0133 |
>
> * [C1] https://en.wikipedia.org/wiki/Numerical_differentiation

---

> ### Author Response · Authors · 2023-11-20
> **Response to Reviewer cbcH (Part 2/2)**
>
> **W3. Provide other metrics that can show the correlation between human preference and CAS in modalities experiments. (Appendix B.2)**
>
> |                                             | CAS(0~25%) | CAS(25~50%) | CAS(50~75%) | CAS(75~100%) |
> |---------------------------------------------|------------|-------------|-------------|--------------|
> | Avg. Human Score in InstructPix2Pix         | 0.341      | 0.489       | 0.568       | 0.6          |
> | Avg. Human Score in ControlNet (Canny Edge) | 0.4875     | 0.4875      | 0.5         | 0.525        |
> | Avg. Human Score in ControlNet (Scribble)   | 0.425      | 0.45        | 0.55        | 0.575        |
> | Avg. Human Score in AudioLDM                | 0.4        | 0.439       | 0.575       | 0.585        |
>
> *Note: CAS(0 ~ 25%) represents the lower 25% of CAS scores when histogram is divided; CAS(25 ~ 50%), CAS(50 ~ 75%), and CAS(75 ~ 100%) represent the 25 ~ 50%, 50 ~ 75%, and top 25% percentiles, respectively.*
>
> To recap, in Sec. 5.2, the performance of CAS across various modalities was measured under the preference of five participants for each test case. Therefore, there are average human scores for each image. Based on this, we have newly analyzed the correlation between CAS and average human scores. We divided the CAS scores into histogram intervals based on percentiles and measured the corresponding average human score. As shown in the table, there is a tendency for the average human score to increase with higher CAS scores, indicating a correlation between CAS and human preference.
>
> We have newly added this result in Appendix B.2.
>
> **Q1. Will the dataset of human evaluations (used in Sec. 5.2) be released along with the code?**
>
> We will! Thanks for your interest.

---

> > ### Comment · Reviewer_cbcH · 2023-11-20
> >
> > Thanks for the detailed reply and additional results. I have updated my recommendation to an accept (8).

---

> > > ### Author Response · Authors · 2023-11-21
> > > **Thank you**
> > >
> > > Thank you for the quick response and for raising the scores. We are pleased that the additional experiments we conducted seem to have been favorably received by the reviewer. Additionally, the experiments requested by the reviewer have provided us with an opportunity to gain a deeper understanding of our technique. Thank you.

---

### Official Review · Reviewer_uTB1 · 2023-10-30

**Soundness:** 4 excellent
**Presentation:** 3 good
**Contribution:** 4 excellent
**Rating:** 8
**Confidence:** 4

**Summary:**

In this paper authors design a novel metric to decide whether result of conditional sampling from diffusion model should be rejected or accepted. Noting that typically user is presented a grid of images and user has to do the wotk himself. Obviously, it would be beneficial to automatize this process. The key idea is to accurately compute difference of unconditional  and conditional log probability. And the key difference to literature is that no curated dataset is needed and separately trained  scoring model.

**Strengths:**

I really liked this paper, it presents an interesting problem and a narrative on how the problemd was solved.  And the idea also makes sense, how to improve the log p(x) computation is reasonable, I also liked solution to compute the difference in N(0,I) and then correct the difference via computing the path.

**Weaknesses:**

Two biggest issues to me are in empirical validation and lack of complete derivations of the main results.
- Main issue with the empirical results is that authors report accuracies. But the accuracy is notoriously bad measure for binary classification case, such as this (keep image or rejecct it). Reasonable metrics are for example, equal error rate (EER) or AUROC. I personally like EER. EER can be nicely computed for example with (https://github.com/bsxfan/PYLLR). Interestingly, same package also allows to plot DET plot that shows the whole ROC in a bit more meaningful way.
- I think it i would be especially useful to have the complete derivations of the main result in the appendix.
- Eq. (9)  also needs a bit more explanation that could be written in the Appendix.

- Some notatioanl issues, such as vectors are marked with bold symbol and non-bold symbol.
- What is the intuition that in Fig 2, - log p(x) shows better performance than log p(x)?
- How was ground truth estimated from Van Gogh dataset? In Pick Score dataset I see that human annotation is used. Did you annotate yourself Van Gogh dataset?

**Questions:**

- About Table 1, I wonder why RMSE is biggest at the edges (\sigma = 01, and \sigma = 0.00001) ?

---

> ### Author Response · Authors · 2023-11-20
> **Response to Reviewer uTB1 (Part 1/2)**
>
> We thank the reviewer for the interest in our work and the valuable feedback that strengthens our paper. The segments of the revised paper that reflect the received feedback are marked in green color.
>
> **W1. Report an Equal Error Rate (EER) for the Pick Score dataset experiment.**
>
> Note that measuring EER for the experiment in the Pick Score dataset is non-trivial. To recap, each test case of the Pick Score dataset involves a pair of images generated from the same prompt and the label, which image is preferred by the human. Then, each T2I alignment score was measured for these images, and we measured whether the score of the human-preferred image is higher than the other one or not. We reported this as accuracy in the experiment. This is markedly different from binary classification, which involves determining whether a single image is positive or negative. This discrepancy makes it hard to measure EER directly in the provided experiment.
>
> Nevertheless, we have measured EER as requested, albeit with a slight modification. First, suppose there is a pair of images (image 1, image 2) generated for a single prompt, and there is a human preference for one of these images. We set the answer as 1 for the preferred image and 0 for the less preferred one. Next, we set the prediction of each score for image 1 as (score(image 1) - score(image 2)) and the prediction for image 2 as (score(image 2) - score(image 1)). After setting up answers and predictions for each image in this manner, we measured the EER.
>
> | Method   | CLIP Score | Image Reward | HPS   | Pick Score | Ours  |
> |----------|------------|--------------|-------|------------|-------|
> | Accuracy | 0.580      | 0.621        | 0.697 | 0.721      | 0.622 |
> | EER      | 0.419      | 0.383        | 0.306 | 0.278      | 0.376 |
>
> As shown in the table, EER measured by this scheme follows the behavior of accuracy. We have added this new result in Appendix B.1. Thanks for suggesting this experiment, which improves the evidence of our claim in this work.
>
> **W2. Provide the complete derivations of main results (CAS) and more explanation of Equation 9 in the Appendix.**
>
> We have newly added full derivations of CAS and more explanation of Equation 9 in Appendix D.1/D.2.
>
> **W3. Notational issues. Vectors should be marked with bold symbols.**
>
> We have reflected all the comments in the revised paper.
>
> **W4. What is the intuition that in Fig. 2,** $-\log p_\mathbf{\theta}(\mathbf{x})$ **shows better performance than** $\log p_\mathbf{\theta}(\mathbf{x})$**?**
>
> As stated in the main paper, we believe that the conditions, which users typically desire to generate images for, are likely to be unobserved in the training distribution. Therefore, our hypothesis is that if a generated image complies with the input condition well, it may turn out a sample from out-of-distribution in terms of the diffusion model's training data, so that the low value of $\log p_\mathbf{\theta}(\mathbf{x})$ is measured. The results in Fig.2, which shows $-\log p_\mathbf{\theta}(\mathbf{x})$ tends to perform better than $\log p_\mathbf{\theta}(\mathbf{x})$, may hint to support this hypothesis.
>
> This observation is important as it underpins our rationale for debiasing $\log p_\mathbf{\theta}(\mathbf{x})$ from $\log p_\mathbf{\theta}(\mathbf{x}|\mathbf{c})$. The second observation in Fig. 2 was that $\log p_\mathbf{\theta}(\mathbf{x}|\mathbf{c})$ is highly biased to $\log p_\mathbf{\theta}(\mathbf{x})$. Therefore, $\log p_\mathbf{\theta}(\mathbf{x})$, which shows a lower value for images well aligned with the condition, degrades the effectiveness of $\log p_\mathbf{\theta}(\mathbf{x}|\mathbf{c})$ as a measurement for alignment score.
>
> This phenomenon is detailed in the response to the W1 asked by Reviewer cbcH with additional toy experiments, which we recommend the reviewer to check it. With newly conducted toy experiments, we have shown that 1) since the user-specified condition is usually not observed in the training procedure of diffusion, a generated image $\mathbf{x}$ complying with the given condition is actually often out-of-distribution in the training distribution, resulting in a lower measured $p_\mathbf{\theta}(\mathbf{x})$ , 2) this effect is so strong that it biases $p_\mathbf{\theta}(\mathbf{x}|\mathbf{c})$ , and 3) by debiasing the conditional probability with the probability, i.e., $CAS(\mathbf{x},\mathbf{c},\theta)=\log p_\mathbf{\theta}(\mathbf{x}|\mathbf{c}) - \log p_\mathbf{\theta}(\mathbf{x})$ , it facilitates to measure alignment between image and conditions.

---

> ### Author Response · Authors · 2023-11-20
> **Response to Reviewer uTB1 (Part 2/2)**
>
> **W5. How was ground truth estimated from Van Gogh dataset?**
>
> As stated in “Experiment setting in Van Gogh dataset” of Sec. 5.1, there are two types of generated images: 1) the images generated directly from the “Van Gogh style, {additional prompts}” themselves, and 2) the images generated from same prompts but with de-weighted Van Gogh style using diffusers module [C1]. We designated the ground truth for the first group as Van Gogh and for the second group as non-Van Gogh. Examples of this can be found in Fig. 5, where all samples are clearly distinguished. Additionally, we conducted a human verification step.
>
> We have detailed this implementation in Sec. 5.1 of the revised paper. We will also provide the dataset if accepted.
>
> - [C1] https://huggingface.co/docs/diffusers/using-diffusers/weighted_prompts
>
> **Q1. In Table 1, why NRMSE is the biggest at the edge ($\sigma=10^{-1}$, $\sigma=10^{-7}$)?**
>
> The sigma is a hyper-parameter that controls the approximation quality of gradient by numerical differentiation. According to the definition of the numerical differentiation, it is obvious for Normalized Root Mean Square Error (NRMSE) to be large when $\sigma=10^{-1}$.
>
> However, the reason for the high value of NRMSE when $\sigma=10^{-7}$ is different, because the NRMSE should decrease, as $\sigma$ approaches zero. It has been well-known that with the limited precision number like float32 we used, when $\sigma$ is set to be too low, the numerical differentiation causes significant rounding errors, leading to the gradient being calculated as zero [C2]. Thus, the widely accepted and common practice is $\sigma=10^{-3}$ (refer to [C2]), which is consistent with our numerical analysis. We have newly added this phenomenon with a more detailed explanation in Appendix A.3 with a newly attached histogram in Fig. 7.
>
> Furthermore, the table below shows the NRMSE between log probability computed by backpropagation and our approximation with various $\sigma$ for the Dreamlike Photoreal 2.0 model (Stable Diffusion v1.5 was employed in the NRMSE table in the main paper.) It can be observed that the approximation reaches the best performance when $\sigma=10^{-3}$. The results also support that practitioners can set $\sigma=10^{-3}$ regardless of the model. We have newly included these experiments in Appendix A.3 with additional details.
>
> | $\sigma$ | $10^{-1}$  | $10^{-2}$  | $10^{-3}$  | $10^{-4}$  | $10^{-5}$  | $10^{-6}$  | $10^{-7}$  |
> |--------|--------|--------|--------|--------|--------|--------|--------|
> | NRMSE  | 0.3048 | 0.0529 | 0.0051 | 0.0081 | 0.1983 | 0.6777 | 8.0133 |
>
> * [C2] https://en.wikipedia.org/wiki/Numerical_differentiation

---

> ### Author Response · Authors · 2023-11-21
>
> If there are any additional discussion points or questions, we are happy to discuss. Thank you.

---

> > ### Comment · Reviewer_uTB1 · 2023-11-22
> > **Reviewer answer**
> >
> > Thanks a lot for a diligent rebuttal. You have answered all my concerns and questions in a thorough way.
> >
> > I will raise my score.

---

### Official Review · Reviewer_8WfE · 2023-11-03

**Soundness:** 3 good
**Presentation:** 2 fair
**Contribution:** 3 good
**Rating:** 8
**Confidence:** 4

**Summary:**

This paper introduces a novel methodology to calculate a condition alignment score (CAS), which is both training-free and applicable across various modalities. The proposed approach involves calculating the ranking metric $\log p_\theta(x|c) - \log p_\theta(x)$ to serve as the alignment score. The authors leverage the probability estimation from flow ODEs to determine the requisite probabilities (with a trained diffusion model (DDIM)). Empirical findings support the efficacy of the CAS method.

**Strengths:**

- The CAS method's universality across different modalities and its training-free nature are significant advantages.
- The proposed framework is principled and reasonable.

**Weaknesses:**

- This paper can be improved by adding more essential technical details. For example,
  - This paper follows many prior works on diffusion model, so there are a lot of citations. When referring to a cited paper, it's unclear for the readers what the paper is about. Maybe a small summary or referring by their method names will be better.
  - The captions can be improved. For example, consider adding a detailed description for the piepline of the method in Figure 3. Explain the full name and formula of NRMSE in Table 1.
  - More ranking examples can be shown in Appendix.
- The experiments can be improved in those aspects:
  - Report the standard deviation for experiments. Each prompt used in Van Gogh Dataset can be used to generated more groups of test samples to test the baselines.
  - The numbers of test samples in different experiments are small. As the accuracy reported in Table 3 is near to 50%, it's hard to judge whether this method works well.
  - The ablation on $\lambda$ for the ranking metric of CAS is missing.
- It will be insightful to show some failure cases and analyze the failure mode to help understand the limitations of CAS.
- There are some typos, for example, the caption for Table 1 is ended without any punctuation.

This paper can be greatly improved after a careful revision. The writing for the current version is a bit rough and obscure. Many details should be added.

**Questions:**

- Could you clarify how the probabilities in Figure 2 were calculated? Are they derived using Equation 4 in conjunction with the Skilling-Hutchinson trace estimator?
- Has the DDIM Recursive Inversion been introduced in previous literature?
- The poor performance of all baselines on the Van Gogh dataset is unexpected, especially considering that the sample in Figure 5 appears distinguishable. Could you provide an explanation for this phenomenon?

---

> ### Author Response · Authors · 2023-11-20
> **Response to Reviewer 8WfE (Part 1/2)**
>
> We thank the reviewer for the interest in our work and the valuable feedback that strengthens our paper. The segments of the revised paper that reflect the received feedback are marked in red color.
>
> **W1-1. When referring to a cited paper, provide a small summary or refer by their method names.**
>
> As requested, we have partially reflected this in Sec. 2 (Related Work). We already referred the prior works in the experiment section by their names in the initial submission.
>
> However, if the request was to explain or name the ScoreSDE [C1] paper specifically, we intentionally avoided doing so to prevent misunderstanding. Although we cited the ScoreSDE paper multiple times for the idea of probability computation, the core of the ScoreSDE paper and probability computation are almost unrelated. The referred probability computation can be found at the end of the appendix of the ScoreSDE paper. Therefore, instead of explicitly referring to the ScoreSDE paper by its method name, we described it as "the method proposed by Song et al." If there are any references we might have missed that could aid understanding, we would appreciate the advice.
>
> * [C1] Song et al., “Score-based generative modeling through stochastic differential equations.” ICLR, 2021.
>
> **W1-2. The captions can be improved, e.g., Fig. 3 and Table 1.**
>
> We have revised captions (Fig. 3, Table 1, and Table 2) in this revision. Thanks.
>
> **W1-3. Provide more ranking examples in Appendix.**
>
> As requested, we have added more ranking examples in Appendix E of this revision.
>
> **W2-1. Report the standard deviation for experiments in Pick Score dataset and Van Gogh dataset. (Table 2)**
>
> As requested, we have repeated the experiment in Sec. 5.1 five times to calculate the standard deviation in Table 2 of the revised paper. On the Pick Score dataset, CAS with DDIM inversion had an average accuracy of $0.593$ (std: $0.004$), while ours with the 2nd order recursive inversion yields $0.622$ (std: $0.005$). On the Van Gogh dataset, the average accuracy for CAS using DDIM inversion was $0.401$ (std: $0.033$), and with the 2nd order recursive inversion, it was $0.412$ (std: $0.013$). These results indicate our technique's higher and stabler performance and show the effectiveness of our 2nd order recursive inversion.
>
> Regarding the results of the 2nd order recursive inversion, to apparently present the statistical significance of our results, we have added them as confidence interval in Table 2 of the revised paper. With 95% confidence interval, the accuracy is $0.622 \pm 0.004$ on the Pick Score dataset, and  $0.412 \pm 0.011$ on the Van Gogh dataset.
>
> **W2-2. The number of test samples is small in Table 3 (other modalities experiment).**
>
> As requested, we have re-experimented Table 3 by adding additional 80 test samples to the test set of each modality and by measuring accuracy on the new test set. The results are reported in Table 3 of the revised paper, which shows the accuracy of CAS is around or higher than 0.6 in most of the modalities.
> Note that the accuracy for CAS on each modality is based on 5 participants for 180 samples now, which deems sufficient. We have newly reported the p-values as well, which imply the probability of obtaining results of other methods better than our accuracy when participants make random 50:50 choices. The measured p-value is $1.74\cdot10^{-8}$ for InstructPix2Pix, $3.18\cdot10^{-18}$ for ControlNet (Canny Edge), $2.16\cdot10^{-11}$ for ControlNet (Scribble), and $3.22\cdot10^{-7}$ for AudioLDM, demonstrating the **statistical significance of our results**. We have newly included this result in Appendix B.2.
>
> Also, note that our evaluation is not entirely based on human studies. In fact, most of our evaluations are quantitative and extensive, which are sufficient to support our contribution and findings across our paper. Through this rebuttal, we have added more results requested by the reviewers, which improve our work meaningfully. Thanks for the suggestion.
>
> **W2-3. The ablation on $\lambda$ (Appendix C.1)**
>
> As requested, we have newly conducted the ablation study of the accuracy of CAS according to $\lambda$ on the Pick Score dataset, and have added this result with discussions in Appendix C.1 of this revision.
> | $\lambda$ | 0.8 | 0.9 | 1.0 | 1.1 |
> | --- | --- | --- | --- | --- |
> | Acc. on Pick Score test set | 0.532 | 0.525 | 0.622 | 0.494 |
>
> We found that $\lambda$ is optimal at 1, which directly measures the ratio of two probability $\log (p_\mathbf{\theta}(\mathbf{x}|\mathbf{c})/p_\mathbf{\theta}(\mathbf{x}))$. We postulate that this balance is favored, because of the quantity balance between $\log p_\mathbf{\theta}(\mathbf{x}|\mathbf{c})$ and $\log p_\mathbf{\theta}(\mathbf{x})$, which is discussed with toy experiments in the response to the question W1 by Reviewer cbcH. We invite the reviewer to also check the response to W1 of Reviewer cbcH.

---

> ### Author Response · Authors · 2023-11-20
> **Response to Reviewer 8WfE (Part 2/2)**
>
> **W3. Provide the limitations of CAS.**
>
> After receiving a request about limitations, we have conducted additional analysis and have newly discovered a new interesting finding that the performance of CAS decreases when comparing images sourced from different generative models.
>
> Specifically, the HPS dataset, a T2I alignment score benchmark, includes images synthesized by 20 different models, and our technique showed lower accuracy compared to the baseline on this dataset. We have revealed that this limitation stems from a new interesting bias of CAS. CAS is measured higher for images that are generated from the same diffusion model on which CAS is computed on, while leading to lower accuracy for those generated from other generative models.
>
> We see this intriguing property as a favorable property, rather than just a weakness. Thereby, we advantageously leverage and extend this characteristic to the other challenging applications of fake detection and image generation source detection, of which details and further discussion can be found in the response to Q3.
>
> **W4. Typos.**
>
> We have polished and fixed some typos. We will further polish in the camera ready. Thanks.
>
> **Q1. Could you clarify How the probabilities in Fig. 2 were calculated? Are they derived using Equation 4 in conjunction with the Skilling-Hutchinson trace estimator?**
>
> Yes. As mentioned, they are derived using Equation 4 in conjunction with the Skilling-Hutchinson trace estimator.
>
> **Q2. Has the DDIM Recursive Inversion been introduced in previous literature?**
>
> To the best of our knowledge, our paper is the first to introduce DDIM recursive inversion. Please inform us if there is any other related work that the reviewer thinks we need to add.
>
> **Q3. The unexpected poor performance of baselines on the Van Gogh dataset – Why CAS perform best in the Van Gogh dataset? (Appendix A.2)**
>
> To address the question, we have investigated the training set of ImageReward. We found that out of the total 9K training prompts, only 21 are related to Van Gogh, which we suspect contributes to the lower performance. This is a common issue shared by all learning-based metrics; they tend to underperform on data they have not frequently encountered. Despite this, the superior performance of CAS is intriguing, and we delve into analyzing it.
>
> This hints to us that our CAS is sensitive to out-of-distribution cases. By leveraging this observation, we have newly revealed **the
> strong capability of CAS to detect out-of-distribution samples**. For the analysis, we have generated 200 images from each of the three different diffusion models. We have also prepared 200 real images. Then, we measured the average CAS of each group of images according to the diffusion models for image generation, and CAS is computed with respective diffusion models that is used for generating each group of images. The following table presents the results of the experiment.
>
> | Model \ Image Source    | Dreamlike Photoreal (DP) | Open Journey (OJ) | Stable Diffusion (SD) | Real  |
> |-------------------------|--------------------------|-------------------|-----------------------|-------|
> | Avg. CAS measured by DP model | 189.90                   | 140.88            | 32.39                 | 10.47 |
> | Avg. CAS measured by OJ model | 121.80                   | 171.81            | 51.04                 | 27.85 |
> | Avg. CAS measured by SD model | 72.50                    | 54.13             | 163.24                | 36.79 |
>
> As shown in the above table, CAS is notably higher for images generated by the same diffusion model that CAS employs. The characteristic of CAS producing high values for in-distribution samples and low values can be straightforwardly applied to **fake image detection** and **image generation source model detection** tasks.
>
> We simply implement the detection methods as follows: 1) we compute CAS values from various diffusion models for an input sample, 2) we use the computed CAS values as an input feature vector, and 3) we train a MLP layer with a supervised dataset of respective tasks.
>
> For the fake image detection application, the accuracy of 0.92 was measured, outperforming the current SOTA fake detection model [C2] whose accuracy is 0.87, trained on the same data. The remarkable point here is that diffusion models leveraged to generate samples in the test data differ from those in train data.
>
> For the source model detection, we reached an accuracy of 0.90.
>
> More detailed results have newly included in Appendix A.2 of the revised paper. Our future work is to further refine this technique (applying MLP to CAS from various models) to better utilize it in fake detection and image generation source detection.
>
> * [C2] Ojha et al., “Towards universal fake image detectors that generalize across generative models.” CVPR, 2023.

---

> > ### Comment · Reviewer_8WfE · 2023-11-21
> >
> > Dear Authors,
> >
> > Thank you for your detailed replies. I find your paper to be interesting. I think the methodology you've introduced may offer some insights for various applications. Given this, I have decided to increase my evaluation to a score of 8.
> >
> > Best regards,
> >
> > Reviewer 8WfE

---

> > > ### Author Response · Authors · 2023-11-21
> > > **Thank you**
> > >
> > > Thank you for raising the scores and for your interest in our paper. We also appreciate your assessment that our methodology will provide insights across many fields. Additionally, the experiments requested by the reviewer have further reinforced our confidence in the effectiveness of our method.

---

### Author Response · Authors · 2023-11-21
**Response to all reviewers**

We are grateful for the constructive feedback provided by the reviewers. Our paper provides the Condition Alignment Score (CAS), the universal alignment score measure between conditions and images (or audio) using only the diffusion model, without additional data or training.

The reviewers have recognized our work for targeting a significant problem (```cbcH```, ```uTB1```), its universality across various modalities (```8WfE```, ```cbcH```), and its training-free nature (```8WfE```, ```uTB1```). Specifically, reviewers (```uTB1```, ```cbcH```) acknowledged our recursive DDIM inversion and its use in accurately measuring CAS, and also acknowledged our achievement in speeding up the CAS measurement through approximation. Our paper's clear writing (```chcH```) and reasonable approach to problem-solving (```uTB1```, ```8WfE```) were also noted.

Please note that we have color-coded the revised sections based on the contributing reviewers: modifications suggested by ```8WfE``` are marked in red, those by ```uTB1``` in green, and ```cbcH```'s contributions in blue.

Despite unanimous agreement on our raising of important issues and remarkable experimental results through a logical development, the reviewers asked more analysis of our CAS and additional details of experimental results. In this rebuttal, we have conducted all requested experiments to resolve the reviewer’s concerns, significantly improving our work.

**More analysis of CAS (Each element has been newly added to Appendix A.1, A.2, and A.3, respectively.)**

- Reviewer ```cbcH``` requested additional analysis on the probability measured by diffusion through a toy experiment. In conducting the experiments requested, we have reaffirmed that 1) $p_\mathbf{\theta}(\mathbf{x})$ is measured low for images well-generated from complex conditions, 2) $p_\mathbf{\theta}(\mathbf{x}|\mathbf{c})$ is strongly biased by this $p_\mathbf{\theta}(\mathbf{x})$, and 3) CAS, which debiases $p_\mathbf{\theta}(\mathbf{x})$ from $p_\mathbf{\theta}(\mathbf{x}|\mathbf{c})$, effectively measures the alignment between images and their conditions.
- Reviewer ```8WfE``` queried why other metrics underperformed on the Van Gogh dataset. We first confirmed that Van Gogh data is a rare case in the training set of these metrics. This highlights our claim that learning-based scores struggle to capture styles from specifically finetuned diffusion models. We have further analyzed the reason for our superior performance in Van Gogh dataset and **discovered our method's strong capability to detect out-of-distribution samples**. By applying this, **we surpassed the fake detection SOTA model's performance** in a toy experiment using only CAS and MLP.
- Reviewer ```cbcH``` and ```uTB1``` requested an analysis on why NRMSE increases with smaller $\sigma$ values in Table 1. Our new experiments have confirmed this as a float32 precision issue, demonstrating the appropriateness of our sigma setting.

**Additional experimental results.**

- T2I alignment score experiments: We have repeated the experiment five times and reported the accuracy with 95% confidence interval (```8WfE```/Table 2). We have also measured a slightly modified version of the Equal Error Rate (EER) (```uTB1```/Appendix B.1).
- Other modalities experiments: We have added an additional 80 test samples to the test set of each modality and also measured the p-value to show our technique’s reliability.  We have additionally measured the mean human score with respect to histograms divided by CAS, examining the correlation between our metric and human preference (```cbcH```/Appendix B.2). We have added the ranking examples (```8WfE```/Appendix E).

These experiments align with our original findings, bolstering our technique's reliability.

We have reflected all results and minor details (Appendix C: $\lambda$ ablation studies / Appendix D: detailed derivations / Overall: bold and non-bald symbols, typos, captions, etc.) in the paper.

In conclusion, these extensive additional experiments prove the stable performance of our technique on various modalities, the effective functioning of introduced methods, and hypotheses. All improvements are thanks to the reviewers, for which we are grateful.

---

### Meta-Review · Area_Chair_2GBZ · 2023-12-06

**Metareview:**

The paper proposed the conditional alignment score, which is a novel universal alignment score measure between conditions and the generated outputs, using only the diffusion model and no additional data or training is needed.

Strengths wise, as agreed with all the reviewers, the problem the paper solves is important, the proposed framework is novel, principled and reasonable. Moreover, the universality and training-free nature are significant advantages.

In the discussions, reviewers requested additional technical details, extra experiments and metrics. The authors have addressed them and revised the paper accordingly, which are acknowledged by the reviewers.

**Justification For Why Not Higher Score:**

It's a good paper but not a ground breaking work.

**Justification For Why Not Lower Score:**

The topic is interesting and the proposed solution is novel and experimental validations are solid.

---

### Decision · Program_Chairs · 2024-01-16

Accept (spotlight)